# Thyme: Think Beyond Images

**Yi-Fan Zhang,**[*] **Zhang Zhang,**[†] **Liang Wang**
Institute of Automation, Chinese Academy of Sciences (CASIA)

**Chaoyou Fu**[†]
Nanjing University (NJU)

**Xingyu Lu, Xiao Hu**
Tsinghua University (THU)

**Shukang Yin**
University of Science and Technology of China (USTC)
Wei Chen, Bin Wen,[†] Kaiyu Jiang, Changyi Liu, Tianke Zhang,
Haonan Fan, Kaibing Chen, Jiankang Chen, Haojie Ding, Kaiyu Tang,
Fan Yang, Tingting Gao, Guorui Zhou
Kwai Keye

## Abstract

Following OpenAI's introduction of the "thinking with images" concept, recent efforts have explored stimulating the use of visual information in the reasoning process to enhance model performance in perception and reasoning tasks. However, to the best of our knowledge, no open-source work currently offers a feature set as rich as proprietary models (OpenAI O3 (OpenAI, 2025)), which can perform diverse image manipulations and simultaneously enhance logical reasoning capabilities through code.

In this paper, we make a preliminary attempt in this direction by introducing **Thyme** (**Th**ink Beyond I**mage**s), a novel paradigm for enabling multimodal large language models to transcend existing "think with images" approaches by autonomously generating and executing diverse image processing and computational operations via executable code (Figure 2). This approach not only facilitates a rich, on-the-fly set of image manipulations (e.g., cropping, rotation, contrast enhancement), but also allows for mathematical computations, all while maintaining high autonomy in deciding when and how to apply these operations. We activate this capability through a two-stage training strategy: an initial Supervised Fine-Tuning (SFT) on a curated dataset of 500K samples to teach code generation, followed by a Reinforcement Learning (RL) phase to refine decision-making. For the RL stage, we manually collect and design high-resolution question-answer pairs to increase the learning difficulty, and we propose **GRPO-ATS** (Group Relative Policy Optimization with Adaptive Temperature Sampling), an algorithm that applies distinct temperatures to text and code generation to balance reasoning exploration with code execution precision. We conduct extensive experimental analysis and ablation studies. As shown in Figure 1, comprehensive evaluations on nearly 20 benchmarks show that Thyme yields significant and consistent performance gains, particularly in challenging high-resolution perception and complex reasoning tasks.

## 1 Introduction

Recent years have witnessed considerable interest in enabling multimodal large language models (MLLMs) (Bai et al., 2025a; Team et al., 2025; Yang et al., 2025) to "think with

---

[*]Project Leader
[†]Corresponding Author

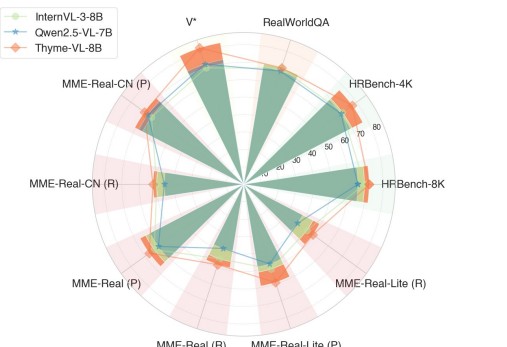 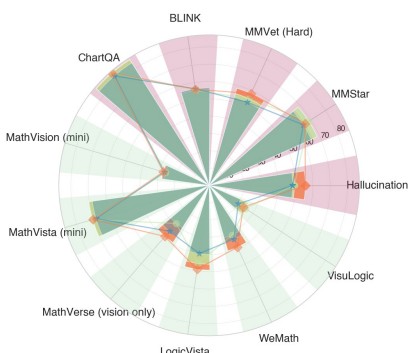

Figure 1: **Benchmark performance of Thyme.** The comprehensive set of image manipulation capabilities enables Thyme to achieve significant improvements over the baseline in perception tasks. By leveraging its ability to convert complex mathematical reasoning into executable code, it consistently outperforms baselines in mathematical reasoning benchmarks. Furthermore, the observed gains across a wide range of general benchmarks further validate the effectiveness of our training approach.

images"—that is, to incorporate visual inputs as part of their reasoning process. Existing approaches to realize this capability generally fall into two categories. One category involves *thinking with generated images* methods (Chern et al., 2025; Zhang et al., 2025a;b), in which the model first generates an image to guide subsequent reasoning. This approach offers significant advantages in scenarios requiring imagination or auxiliary visual cues, such as drawing auxiliary lines to solve mathematical problems. However, generated images often suffer from quality limitations that hinder the preservation of fine-grained original details, thus constraining improvements in core visual perception tasks. Moreover, the substantial computational cost associated with image generation results in considerable inference latency, which undermines real-time applicability. The other category focuses on *thinking with cropping*, in which the model is trained to select relevant image regions (e.g., by outputting bounding boxes (Shen et al., 2024; Zhang et al., 2025c)), which are then cropped and processed externally. While this strategy enhances perception accuracy and mitigates hallucination, its functionality remains relatively limited. Specifically, it can only perform cropping through bounding box outputs, falling short of the multifunctionality demonstrated by OpenAI's O3 (OpenAI, 2025), which supports a broader range of operations such as rotation, contrast enhancement, and even coding.

To overcome these challenges, we propose a novel paradigm called **Th**ink Be**y**ond I**m**ag**e**s (Thyme), illustrated in Figure 2, guided by four core principles:

◇ **Rich Functionality**: Enables the model to support a wide range of common image operations such as cropping, scaling, rotating, and contrast enhancement, as well as complex mathematical computations.
◇ **High Autonomy**: Thyme exhibits high autonomy, capable of deciding whether to perform image operations, determining what operations to apply, and executing the functions by dynamically generating code — all without human intervention for specific tasks.
◇ **Efficient End-to-End Training**: We utilize an end-to-end supervised finetuning + reinforcement learning (SFT+RL) to minimize training costs and rapidly unlock diverse model capabilities. Notably, the SFT stage requires only 200 GPU hours to activate all the functionalities mentioned above, while the RL stage further strengthens these abilities.
◇ **Significant and Stable Performance Gains**: We evaluate nearly 20 benchmarks spanning three major categories: perception, reasoning, and general tasks. Under the Thyme paradigm, the model achieves consistent and substantial performance improvements across all these task categories.

To realize this vision, our technical roadmap comprises the following key components:

⋄ **SFT Data Construction:** Leveraging over 4 million raw data sources, we curate a high-quality SFT dataset of approximately 500K samples encompassing diverse scenarios: (a) image operations and computations requiring no coding; (b) cropping of complex or high-resolution images; (c) correction of images rotated by large angles; (d) enhancement of low-contrast images; (e) tasks involving complex code-based computations to boost understanding; and (f) multi-turn interactive scenarios, such as iterative magnification.

⋄ **SFT Training Strategy:** To accommodate the diversity of functionality, we implement specific training schemes. Specifically, outputs and environment feedback from the sandbox are masked during loss computation to prevent gradient contamination; only the model's last-round response in multi-turn dialogues is used for loss calculation; and annealing techniques are applied on long-tail complex computation samples to thoroughly activate the model's Thyme capabilities.

⋄ **RL Data Construction.** Our RL dataset primarily consists of two parts. The first part involves filtering open-source data, which contains a mixture of perception and reasoning tasks. However, the image resolution and complexity in public datasets are limited. Therefore, we manually collect and annotate 10,000 high-resolution and perceptually challenging complex perception images. These images are carefully curated for high resolution and complexity, focusing on enhancing the model's perceptual capabilities and addressing more difficult tasks.

⋄ **RL Algorithm Design.** During the reinforcement learning phase, we observe that using a high sampling temperature for code generation leads to low code usability—sampling any invalid characters or spaces when generating variable names will cause the entire code snippet to fail execution. To address this, we adopt an adaptive temperature sampling strategy during RL training: we set temperature to 0 for code generation to ensure determinism and execution validity, while using temperature 1 for natural language reasoning to encourage diverse exploration, thereby ensuring code validity and reducing runtime errors.

⋄ **Sandbox Design.** We develop a secure sandbox environment capable of executing model-generated code within strict time limits and returning results. This environment minimizes the model's code burden by automatically handling formatting, variable definitions, and boundary conditions related to image operations, significantly improving code usability without altering its semantic intent.

Thyme effectively fulfills our vision by combining high autonomy with rich functionality. It can assess the complexity of a given problem and determine whether tool usage is necessary. When required, it autonomously defines and invokes tools through code, even performing multiple operations such as cropping, zooming, and rotating in a single execution. Moreover, the SFT training phase requires only about 200 GPU hours to activate the model's fundamental abilities to perform various image manipulations and computations, reflecting favorable computational efficiency. We conduct comprehensive evaluations of Thyme across more than 20 multimodal benchmarks. Experimental results in Figure 1 demonstrate that Thyme exhibits significant and consistent advantages in fundamental perception, complex reasoning, and computation tasks.

## 2 PRELIMINARY AND OVERALL PIPELINE

As shown in Figure 2, the overall pipeline mainly consists of two components: the **model** and the **sandbox**. Given a user input, the model first generates a reasoning process, which includes analyzing the problem and deciding, based on the type and difficulty of the problem, whether to generate code.

If **no code generation is required**, this implies the problem is simple, or that through previous dialogue rounds, relevant code operations have already successfully solved the problem. In such cases, the answer is returned directly. If **code generation is needed**, the model autonomously produces the code. Our training data covers several types of image operations such as *cropping, zooming, rotation, contrast enhancement, and computations*. The model may implement one or a combination of these functions in the generated code, for example performing cropping, zooming, and contrast enhancement simultaneously.

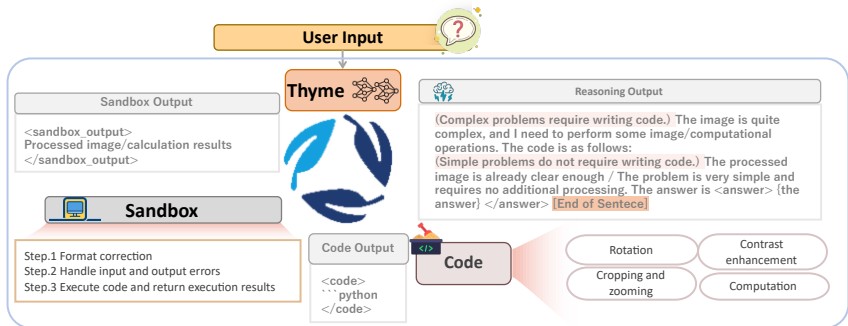

Figure 2: **Overall pipeline of the Thyme**, illustrating the interaction between the model and the sandbox for iterative reasoning and code execution. Key processes such as reasoning, code generation, sandbox execution, and result feedback are highlighted.

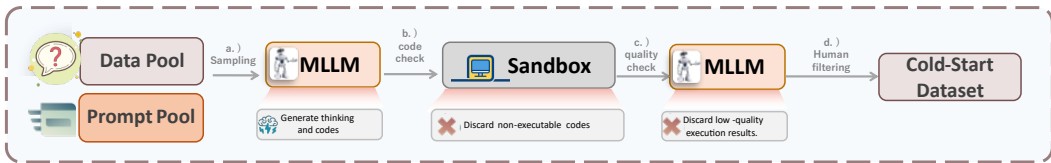

Figure 3: **SFT Data Construction Pipeline.** First, samples are taken from an existing dataset and prompts are constructed based on the target functions (such as cropping, rotating, etc.). The model generates a thinking process and corresponding code based on the prompt. The code is then executed in a sandbox environment to filter out samples that fail to run properly. The remaining samples are reviewed by an additional MLLM to verify whether the code execution results align with the thinking process and effectively answer the question, eliminating invalid code samples. Finally, manual review is conducted to remove low-quality samples, ensuring the quality of the cold-start dataset.

The input parameters of these operations, such as the contrast enhancement factor, crop coordinates, and zoom ratio, are all decided by the model itself with no external constraints.

The generated code is then executed within an external sandbox, whose main function is to securely handle the input code and return execution results. Inside the sandbox, we utilize Python's built-in modules such as `ast`, `autopep8`, etc., to perform code formatting, correct input-output variable properties, and fix code input parameters. This is done without affecting the code functionality but to avoid errors caused by trivial code details, thereby reducing the model's code burden and improving code usability.

Finally, the code execution results are returned to the model for the next dialogue round, and the model conducts reasoning based on these results to continue the interaction. We demonstrate the construction process and considerations for the sandbox section in Section A.

## 3 THYME-SFT COLD START

In this section, we introduce the data construction, data composition, and training strategies employed for the cold start of Thyme.

### 3.1 TRAINING DATA CONSTRUCTION PIPELINE

This subsection details the construction of the training datasets, which are organized into three task categories reflecting increasing complexity. The first category includes tasks suitable for direct answer generation without the need for code. The second category comprises tasks that require image-based operations, such as cropping, rotation, contrast

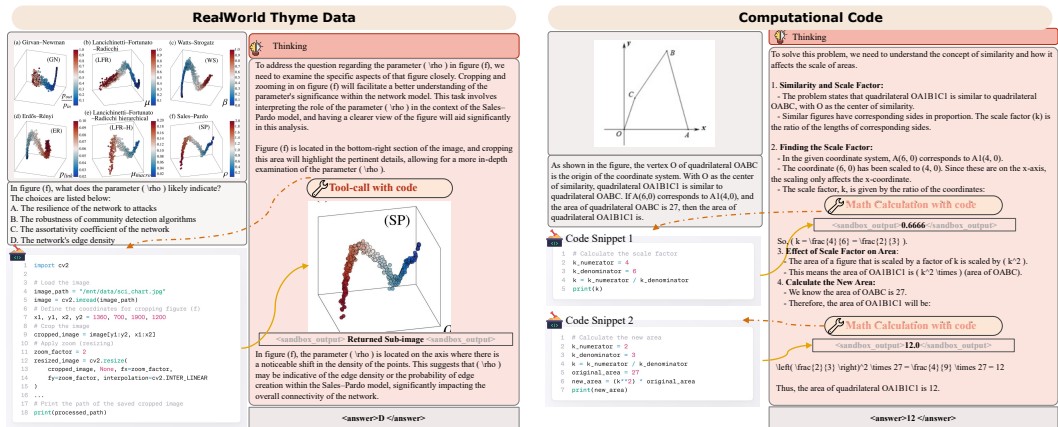

Figure 4: **Visualization of SFT Data instances**. The left side presents a sample of data related to image processing operations, while the right side showcases a sample of data related to complex computations.

enhancement, as well as mathematical computations. Finally, the third category consists of tasks that extend over multiple interaction rounds—for instance, cases where initial image manipulations fail or require refinement, and complex problems that involve multi-turn reasoning and calculations. All original prompts are sourced from a variety of open datasets, including LLaVA-OV-Image (Li et al., 2024a), MM-RLHF (Zhang et al., 2025d), V* (Wu and Xie, 2024), MM-Eureka (Meng et al., 2025), arXivQA (Li et al., 2024b), and Retool (Feng et al., 2025), ensuring broad coverage and diversity. The overall data construction pipeline is shown in Figure 3 and data samples are shown in Figure 4. We defer the detailed data taxonomy and construction methodology to Appendix B.

### 3.2 TRAINING STRATEGY

Our system prompt and user prompt are presented in Table 2 and Table 3, respectively. In the system prompt, we explicitly define the task format, the expected output code style, and the format for sandbox environment execution results, ensuring the model clearly understands the task requirements. Meanwhile, the user prompt includes the image path and image size, facilitating proper image loading and preventing out-of-bounds operations. It also specifies the required output format.

We formalize the training process of Thyme to facilitate a clearer understanding of the employed training techniques. Given an image $I$ and a question $Q$, the model generates a sequence of thinking process $T$ along with optional code $C$. If code is generated, an external sandbox executes it and returns the execution result $S$, which may be one or multiple images, or a computed numeric result. The model iteratively interacts based on the previous execution results, continuing this procedure until the question is resolved and the final answer $a$ is produced. Therefore, each training sample can be represented as:

$$X = \{(I, Q); ([T_0, C_0, S_0], \ldots, [T_t, a])\}$$

where $t$ denotes the maximum number of interaction rounds for the sample.

During training, we encountered several challenges. First, due to the special nature of two-round dialogue data, some unexpected patterns emerged: the model tends to generate an incorrect or insufficient analysis and code in the first round, then corrects it in the second round, rendering the first round essentially ineffective. The second challenge is the relatively small quantity of math data compared to image manipulation data; when trained jointly, the model barely learns to generate computation-related code.

To address these issues, we devise several training-phase strategies:

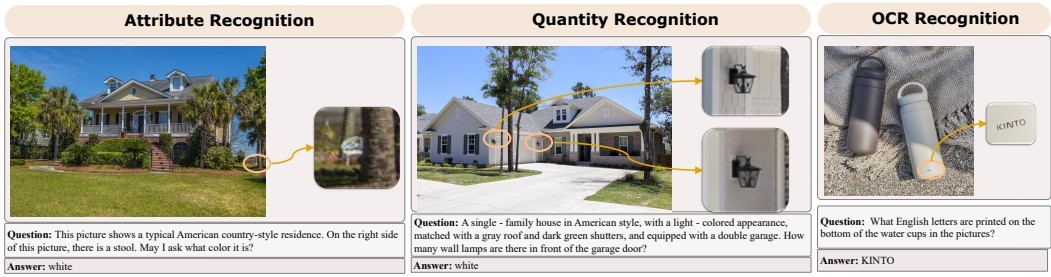

Figure 5: **Visualization of RL Data Instances**. Thyme RL data focuses on complex scenarios and high-resolution image interpretation. Human annotators identify challenging objects within the images, design corresponding questions, and provide answers, along with the appropriate bounding boxes.

◇ **Train on the Last Round Only:** For samples with two or more rounds, mask out all content before and including the last `<sandbox_output></sandbox_output>` tag, such that the model learns only to generate outputs corresponding to the final round. In other words, all $[T_{t-1}, C_{t-1}, S_{t-1}]$ are masked out, and only the logic and results of the last round are learned.

◇ **Sandbox Content Exclusion:** Throughout all training stages, we exclude the sandbox identifier `<sandbox_output></sandbox_output>` markers and sandbox execution results from the training targets. Formally, all $S_{i=1...t}$ are masked out and do not participate in training, thereby preventing the model from learning to directly predict sandbox outputs.

◇ **Math Data Annealing:** To prevent the math data from being overwhelmed by image operation data, first train on all image-related data, then fine-tune exclusively on math data with a lower learning rate. Since math data usually involves multi-turn code with weak inter-round correlations, all rounds are learned jointly in this phase instead of applying the last-round-only strategy.

## 4 THYME-RL

We present the construction of RL data for Thyme, including the selection of data from public datasets and the manual creation of complex real-world visual question answering data to enhance task difficulty. Finally, we introduce the algorithmic and architectural innovations implemented during the RL phase.

### 4.1 RL DATA CONSTRUCTION AND ANNOTATION

**Public Data Selection:** Similar to the existing study (Zheng et al., 2025), we curate RL data from a diverse range of prompt sources, including V* (Wu and Xie, 2024), arXivQA (Li et al., 2024b), and ThinkLite-VL (Wang et al., 2025a), targeting both perception tasks, and reasoning tasks, respectively. Manual verification removes instances where the questions and images do not match or lack relevance.

**Complex Image Data and Question Construction:** Publicly available datasets generally contain images with limited resolution and insufficient perceptual complexity, which substantially restricts the effectiveness of the Thyme framework in reinforcement learning. To address this gap, the following pipeline constructs relevant data:

◇ Manually collect 30,000 high-resolution images from the internet, each with either width or height exceeding 2048 pixels, to ensure sufficient image complexity.
◇ Employ a team of 15 annotators to label these images. The annotation guidelines appear in Section D. Specifically, annotators design, for each image, a concrete question targeting small and challenging-to-recognize objects occupying no more than 5% of the image

resolution, ensuring high recognition difficulty. Users require zooming in on the image to clearly identify these small objects.

All annotation data undergo at least one round of cross-checking, followed by unified inspection and acceptance by three multimodal experts to ensure data quality.

## 4.2 Preliminary of Reinforcement Learning Algorithm

**Training Algorithm:** We adopt a fully on-policy variant of Group Relative Policy Optimization (GRPO) (Shao et al., 2024a) algorithm. For each problem consisting of an image $I$ and a question $Q$, the algorithm samples a group of interaction trajectories $\{\tau_1, \tau_2, \ldots, \tau_G\}$ from the policy $\pi_\theta$, where each trajectory $\tau_i$ represents a complete multi-round interaction:

$$\tau_i = \{(I, Q); ([T_{i,0}, C_{i,0}, S_{i,0}], \ldots, [T_{i,t_i}, a_i])\}$$

The policy is updated by maximizing the following objective:

$$\mathcal{J}_{\text{GRPO}}(\theta) = \mathbb{E}_{(I,Q)\sim D, \{\tau_i\}_{i=1}^G \sim \pi_\theta(\cdot|I,Q)} \left[ \frac{1}{\sum_{i=1}^G |\tau_i|} \sum_{i=1}^G \sum_{j=1}^{|\tau_i|} (A_{i,j} - \beta \mathbb{D}_{\text{KL}}[\pi_\theta||\pi_{\text{ref}}]) \right]$$

where $|\tau_i|$ is the total number of tokens generated by the model in trajectory $i$, specifically: $|\tau_i| = \sum_{k=0}^{t_i} |T_{i,k}| + \sum_{k=0}^{t_i-1} |C_{i,k}| + |a_i|$ and $A_{i,j}$ is the advantage computed from the final rewards $\{r_1, r_2, \ldots, r_G\}$ of the complete trajectories in the same group: $A_{i,j} = \frac{r_i - \text{mean}(\{r_i\}_{i=1}^G)}{\text{std}(\{r_i\}_{i=1}^G)}$ for each token $j$ in $\tau_i$ The term $-\beta \mathbb{D}_{\text{KL}}[\pi_\theta||\pi_{\text{ref}}]$ penalizes large deviations of the current policy $\pi_\theta$ from a reference policy $\pi_{\text{ref}}$, with $\beta$ controlling the strength of the constraint. The reward $r_i$ evaluates the quality of the entire trajectory, including the effectiveness of code generation, sandbox utilization, and the correctness of the final answer $a_i$. Importantly, during the RL stage, the trajectory length $|\tau_i|$ and the summation over tokens in the advantage calculation exclude the sandbox-related content $S_{i,k}$, as these are external observations rather than outputs of the policy model.

## 4.3 Adaptive Temperature Sampling Strategy in GRPO Training

During our preliminary experiments, we observe that code generation is highly sensitive to sampling randomness: even minor stochastic variations (e.g., inserting an extra space in a variable name) often lead to non-executable code. In contrast, natural language reasoning benefits from diverse exploration. Based on this observation, we implement a context-aware temperature scheduling mechanism: whenever the model outputs within a `` block, we fix the sampling temperature to 0; otherwise, we use temperature 1. As illustrated in Figure 6, this is a lightweight, rule-based switch that requires no additional parameters. In our experiments, we observe two main benefits :

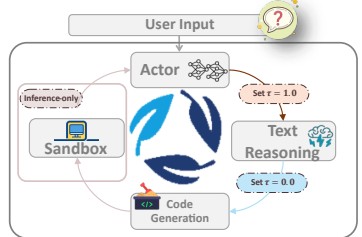

Figure 6: **GRPO-ATS Sampling Pieline.**

◇ It improves sample efficiency. During rollouts of RL, many samples become unusable due to chaotic or erroneous code generation, for example, continuously attempting to generate code incorrectly and failing until the maximum interaction steps are exhausted. Lowering the code temperature helps these samples recover and become valid.

◇ It prevents the model from collapsing into completely avoiding code generation. Since writing code is more error-prone than natural language reasoning, without adjusting the sampling parameters, most code produced during training rollouts is invalid. Over time, this causes the model to become overly conservative and tend to avoid writing code altogether. Reducing the temperature and thus increasing the usability of generated code effectively mitigates this issue.

Table 1: **Performance Comparison on the MME-Realworld.** Thyme-7B demonstrates substantial improvements over the Qwen2.5-VL-7B baseline, particularly in more challenging perception and reasoning domains such as Monitoring and Autonomous Driving, where the baseline's performance is weaker.

| Perception | | | | | |
|---|---|---|---|---|---|
| *Model* | *Monitoring* | *Autonomous Driving* | *OCR* | *Diagram and Table* | *Remote Sensing* | *Overall* |
| Qwen2.5-VL-7B | 38.75 | 22.70 | 87.00 | 78.83 | 45.40 | 60.94 |
| Thyme-VL-7B | 49.27 | 37.46 | 88.36 | 80.86 | 53.93 | 67.10 |
| *Improvement ↑* | *27.14%* | *64.99%* | *1.56%* | *2.57%* | *18.80%* | *10.10%* |
| Reasoning | | | | | |
| *Model* | *Monitoring* | *Autonomous Driving* | *OCR* | *Diagram and Table* | *Remote Sensing* | *Overall* |
| Qwen2.5-VL-7B | 26.10 | 24.25 | 64.80 | 63.40 | - | 38.59 |
| Thyme-VL-7B | 47.39 | 32.29 | 70.60 | 70.40 | - | 48.38 |
| *Improvement ↑* | *81.57%* | *33.16%* | *8.95%* | *11.04%* | *-* | *25.37%* |

**Sampling Optimization:**   Another major challenge stems from the multi-turn interactive nature of the task: the maximum number of interaction rounds imposes a strict sampling time budget, as extended or repetitive dialogues significantly increase computational overhead. When the model enters a cycle of repetitive errors, it produces redundant outputs across multiple turns, thereby wasting computational resources on invalid samples. To mitigate this, we implement an early termination mechanism during training. Specifically, we utilize the Rabin-Karp rolling hash algorithm (Leiserson et al., 1994; Karp and Rabin, 1987) to count occurrences of all substrings of fixed length within the model's output. If any substring repeats so frequently that the cumulative length of its occurrences exceeds 50% of the total output length, we classify the output as repetitive. Once this criterion is met, the sampling process for that particular sample is immediately halted to avoid unnecessary computation. Additionally, during training, we enforce a manual cap on the maximum allowed number of dialogue turns, denoted as `max iterations`. Dialogues that exceed this limit without terminating are truncated to control context length, thereby improving training efficiency.

## 4.4    Reward Function Design:

◇ Formatting Reward: The model's output is required to conform to a strict structure, enclosed by `<think></think>` tags followed by `<answer></answer>` tags. This encourages the model to explicitly perform reasoning before producing a final answer, improving the interpretability.
◇ Result Reward: Since not all targets are strictly numerical or formulaic answers amenable to rule-based comparison, we first apply a rule-based method to directly compare the model output with the ground truth answer. If the rule-based matching fails, we then utilize an auxiliary MLLM, Qwen-2.5-VL-72B, to compare the model-generated output against the ground truth by assessing semantic equivalence and correctness. This hybrid reward approach, combining rule-based and model-based evaluation, improves scoring efficiency and is well-suited for open-ended question answering scenarios.
◇ Consistency Reward (Team et al., 2025; Zhang et al., 2025e): This component evaluates whether the final answer is logically derived from the preceding reasoning steps, ensuring coherence between the thought process and the conclusion.

Integrating the consistency reward alongside formatting and result rewards can cause unintended behavior: the model might receive a high total reward by maintaining consistency even when the chosen answer is incorrect. To prevent this, we define the final reward as follows: Final Reward $r = \text{Result Reward} \times \left(1 + 0.5 \times \text{Consistency Reward}\right) + 0.5 \times$ Formatting Reward. This formulation ensures that the consistency bonus is considered only when the model's answer is correct, avoiding excessive prioritization of consistency at the expense of answer accuracy.

## 5 Experiments

Due to space constraints, we relegate the, experimental setup, baselines and evaluation benchmarks, as well as the ablation studies and case analyses of model outputs to the appendix C. In the main paper, we focus on demonstrating the performance improvements brought by Thyme to the model.

**Main Results:** Figure 1 and Table 4 presents a comprehensive comparison between Thyme and other leading multimodal models across a range of benchmarks covering *Perception*, *Reasoning*, and *General* tasks. In perception tasks, Thyme demonstrates clear advantages even over larger-scale models such as Qwen2.5-VL 32B, indicating that simply scaling model size does not effectively address perception challenges. Instead, Thyme's test-time scaling strategy proves highly beneficial for perception tasks. Furthermore, by converting complex computations into executable code through training, Thyme achieves notable improvements in reasoning abilities. However, in this domain, the benefits of scaling model size are more pronounced, suggesting that reasoning and logical inference capabilities largely depend on the inherent knowledge within the model itself. Thyme mainly enhances visual recognition quality and helps avoid the model independently predicting overly complex computations. Finally, due to the improvements in both perception and reasoning, Thyme shows significant gains in many general tasks, particularly with a substantial reduction in hallucination.

We additionally benchmark Thyme against a broader set of strong baselines, including state-of-the-art commercial APIs and recent tool-use academic methods, on a representative subset of benchmarks. The expanded evaluation setup, conceptual comparisons, and detailed results are provided in Appendix C.4.

**Delve deep into perception tasks:** Taking MME-Realworld as an example, it includes many high-resolution perception tasks in real-world scenarios. In Table 1, we show the performance of Thyme and the baseline on various tasks. It can be seen that on tasks where the baseline model already performs well, such as OCR, Diagram and Table, achieving accuracy of over 60% and even close to 90%, the improvement from Thyme is limited. However, for more difficult tasks where Qwen2.5-VL-7B's perception is relatively poor, such as monitoring and autonomous driving, Thyme shows an improvement of over 25% in both perception and reasoning tasks, with the improvement in reasoning tasks being more pronounced.

We conduct comprehensive ablation studies and extended experiments detailed in Appendix C. These experiments validate key design choices including our SFT training strategies (masking sandbox content, using only final rounds for multi-turn tasks, and math data annealing), our RL reward design (where consistency rewards prove most beneficial). Notably, our approach achieves substantial improvements over concurrent work like Deepeyes-7B and direct RL baselines.

## 6 Conclusion and Limitations

We introduce **Thyme**, a novel paradigm that empowers MLLMs to autonomously generate and execute code for a wide range of image manipulations and complex computations. Our approach utilizes a two-stage training strategy, beginning with a comprehensive SFT phase on a curated 500K-sample dataset to instill foundational coding abilities for tasks like cropping, rotation, and mathematical calculations. This is followed by a RL phase, where we employ our proposed **GRPO-ATS** algorithm to refine the model's decision-making and executional precision. GRPO-ATS adaptively uses different sampling temperatures for text and code generation, which effectively balances creative reasoning with the need for accurate, executable code. Extensive evaluations across nearly 20 benchmarks demonstrate that Thyme achieves significant and consistent performance improvements over the baseline, particularly in challenging high-resolution perception and complex reasoning tasks.

Our work has several limitations that open avenues for future research: 1. **Model Capability.** Thyme's performance is limited by its base model's capabilities in object localization and code generation, occasionally resulting in incorrect cropping or unexecutable code. Stronger foundation models could significantly improve reliability and accuracy. 2. **Evaluation Scope.**

Current benchmarks primarily contain high-quality images from everyday scenarios, lacking robust evaluation for advanced operations like rotation correction or contrast enhancement. Specialized benchmarks for these capabilities would enable more comprehensive assessment.

**Reproducibility Statement.** To ensure reproducibility of Thyme, we provide detailed implementation specifications throughout this work. The two-stage training pipeline (SFT followed by RL) is described in Sections 3 and 4, with specific hyperparameters and training strategies documented. The SFT data construction methodology and sample curation process are detailed in Section 3.1 and Appendix A. Our novel GRPO-ATS algorithm with adaptive temperature sampling is fully specified in Section 4.3, including temperature settings (0 for code generation, 1.0 for text reasoning). The sandbox environment design and security constraints are outlined in Section 2 and Appendix B. Evaluation protocols across nearly 20 benchmarks are standardized and documented in Section 5 and Appendix C. All datasets used in training and evaluation are publicly available with provided references. The reward function components (formatting, result, and consistency rewards) are mathematically defined in Section 4.4. Code implementation and model checkpoints will be released to facilitate the reproduction of our results.

## LLM Usage Declaration

In this research, LLMs were used exclusively for grammar checking and to assist with the clarity of language. No LLM was involved in the ideation or content generation processes. The authors take full responsibility for all content presented in the paper, including any generated by the LLM. We have ensured that the use of LLMs complies with ethical standards and does not constitute any form of scientific misconduct or plagiarism.

## Acknowledgement

This work was jointly supported by the National Key R&D Program of China (Grant No. 2022ZD0117901), the National Natural Science Foundation of China (Grant Nos. 62506158, 62441234, 62322607, 62276261, 62373355, 62306311, and 62236010), the Basic Research Program of Jiangsu (Grant No. BK20251183), and the Beijing Natural Science Foundation (Grant No. L252033).

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

# A  Sandbox Building

Our framework requires a robust sandbox with two essential properties. First, it ensures basic security and robustness for the operating system, guaranteeing that any image processing or computational operations executed within the sandbox complete within a reasonable timeframe or throw appropriate errors. At the same time, it prevents unauthorized system modifications, such as file deletion or renaming, which could corrupt existing data or runtime configurations.

We observe that smaller models (e.g., 7B parameters) often struggle to generate flawless code, frequently encountering minor issues such as formatting problems (improper indentation), boundary condition errors (cropping boxes exceeding image dimensions), and input/output handling (omitting input variables or output printing). Therefore, the sandbox design aims to reduce the model's coding burden by automatically handling these minor issues when possible.

**To address sandbox security and robustness**, we apply the following strategies:

⋄ We define a list of dangerous operations—such as `remove`, `unlink`, `move`, and `rename`—that may cause system data or configuration failures. Before execution, the code is scanned for these operations; if it contains any, execution skips and a warning raises as an error.
⋄ We limit the maximum execution time, empirically set to 10 seconds. Exceeding this limit results in a timeout error. These measures help ensure code isolates from the local system and runs safely.

**To reduce the model's code generation burden,** we implement the following optimizations:

⋄ We apply Python's built-in `autopep8` for code formatting and indentation alignment.
⋄ Using the `ast` module, we traverse the code to identify variables in tuple formats. If variable names contain `box` or `coord`, we infer they represent cropping parameters; these parameters adjust to fit within the input image boundaries to avoid execution errors.
⋄ Before execution, we preset local variables such as `image_path` and import packages like `cv2`. After execution, we check for newly created variables indicative of processed outputs (e.g., those containing `'processed_path'`) to ensure correct execution even if input/output variables do not explicitly define in the model-generated code.
⋄ Additionally, we observe that when the model outputs multiple code segments, context or variable dependencies often arise; for example, certain Python packages are imported only in the first segment, while subsequent segments rely on them directly. To address this, we record all variables throughout the model's code execution process and incorporate historical context in multi-round sandbox invocations.

# B  Long-CoT Data Construction

**Focusing first on the simplest category**, which involves no code generation, the objective is to train the model to confidently bypass the intermediate coding step and directly produce accurate final answers for relatively straightforward questions. These samples derive from LLaVA-OV-Image data, where for each given question and accompanying image, Qwen2.5-VL-72B assesses whether code generation is necessary. The prompt template guiding Qwen2.5-VL-72B's decision-making process is illustrated in Table 9. From the entire pool of samples requiring no tool intervention, we randomly select 100k samples to constitute this training subset.

**Real-World Thyme Data:** Moving towards increased complexity, samples requiring code-based operations undergo rigorous verification and quality control. Initially, all code snippets pass through our sandbox environment where non-executable codes are discarded to ensure only valid samples proceed. Subsequently, the execution outputs, alongside the model's own analysis and answers, feed back into Qwen2.5-VL-72B to verify whether the intended goals—such as clearer identification of the target region by cropping or enhanced readability

through rotation—are effectively achieved. The model flags and eliminates any samples where code does not fulfill those expectations. To further enhance data reliability, two multimodal experts perform an additional round of meticulous screening, filtering out ambiguous or borderline cases beyond the model's discernment. These samples reflect realistic and complex visual-question answering scenarios drawn from more than 4 million instances across public datasets, culminating in a refined collection of 50k high-quality, executable QA pairs.

**Manually Constructed Thyme Data:** Given that real-world Thyme data often have high complexity and quality but limited quantity—we complement this by deliberately creating an additional manually constructed dataset[1]. This curated data supplements and enriches the training distribution, helping the model perform robustly across various task types. The verification and filtering protocols for this synthetic dataset parallel those used for real-world data, ensuring consistent quality standards.

⋄ **Coding Data 1: Cropping** — This dataset draws primarily from V*, which features questions paired with images, corresponding answers, and ground truth bounding box coordinates for target objects. By applying the prompt outlined in Table 10, the model is prompted to explain the rationale behind cropping the object and its contribution to problem-solving. Cropping then proceeds based on the ground truth coordinates before providing answers. Despite leveraging accurate bounding box information, code generation quality constraints restrict the final dataset size to 28k samples.

⋄ **Coding Data 2: Rotation** — Addressing the scarcity of rotated image samples in conventional training datasets, we augment the data by randomly rotating images within the collected pool at angles ranging from 30° to 335°. Applying the prompt template in Table 11, the model generates appropriate processing code and accompanying analytical commentary informed by ground truth rotation angles. Notably, we observe the relative insensitivity of existing MLLMs to rotation angles; that is, without ground truth inputs, the majority of samples cannot be reoriented correctly. After careful filtering, this yields a set of 14k high-fidelity samples.

⋄ **Coding Data 3: Low-Contrast Enhancement** — Derived mainly from OCR data within LLaVA-OV-Image, this subset targets scenarios in which contrast enhancement reduces the difficulty of text recognition, thereby aiding the model's text extraction capabilities. We repurpose prompts akin to those in Table 10, swapping out cropping directives for contrast enhancement instructions. Qwen2.5-VL-72B generates explanations of the enhancement's purpose, corresponding code, and final visual results. The curated dataset contains approximately 10k high-quality training examples.

⋄ **Coding Data 4: Computational Code** — To not only support image manipulation but also foster enhanced logical reasoning through mathematical computation, we construct a specialized math-centric dataset. Its principal source is MM-Eureka and Retool. Initial long-chain-of-thought solutions are generated by Gemini 2.5 Pro, which are then transformed by GPT-4o into executable code segments.[2] This process results in 15k samples focused on computational reasoning via code.

**Multi-Round Conversation:** Except for the Computational Code data, all the aforementioned datasets involve only a single round of code generation and execution. Our experiments reveal that models trained solely on single-turn interactions lack error-correction capabilities. Specifically, even if the model incorrectly crops a sub-region, it proceeds to generate an answer based on the flawed crop without attempting to re-crop or further adjust the targeted area. To address this limitation, we manually construct multi-round conversational datasets aimed at teaching the model to leverage historical code execution outcomes for deeper analysis and error rectification. This multi-round data is categorized into two types:

---

[1]Most real-world data features high-resolution images or complex problems where additional image processing significantly improves perceptual effectiveness. In contrast, manually constructed data often contains smaller resolution images (e.g., V*), where using prompts from Table 9 might cause the model to simply output answers, necessitating specialized construction procedures.

[2]Although Gemini 2.5 Pro produces detailed and comprehensive solutions, its code tends to be overly complex and heavily annotated, making it less suitable for model training. Therefore, GPT-4o handles code conversion to produce leaner implementations.

Table 2: **System Prompt for SFT.**

---

You are a helpful assistant.

Solve the following problem step by step, and optionally write Python code for image manipulation to enhance your reasoning process. The Python code will be executed by an external sandbox, and the processed image or result (wrapped in `<sandbox_output></sandbox_output>`) can be returned to aid your reasoning and help you arrive at the final answer.

**Reasoning & Image Manipulation (Optional but Encouraged):**

- You have the capability to write executable Python code to perform image manipulations (e.g., cropping to a Region of Interest (ROI), resizing, rotation, adjusting contrast) or perform calculation for better reasoning.
- The code will be executed in a secure sandbox, and its output will be provided back to you for further analysis.
- All Python code snippets **must** be wrapped as follows:

  ```
  
  ```python
  # your code.
  ```

  
  ```

- At the end of the code, print the path of the processed image (`processed_path`) or the result for further processing in a sandbox environment.

---

◇ **Further Enhancement:** The model refines or augments previous results based on feedback. We sample data from V* and utilize the prompt shown in Table 10, but initially provide a larger bounding box that fully contains the ground truth cropping box. Based on the code execution results from this first round, the model generates further analysis and code to crop precisely to the ground truth bounding box.

◇ **Error Correction:** The model detects and fixes mistakes identified in earlier steps. Similarly, we provide an incorrect bounding box in the first step. In the second step, the model generates the corrected bounding box with an explanation on why the cropping region needs adjustment.

Such multi-round data effectively instructs the model to revise its prior image manipulation errors and further enhance the quality of its operations.

## C   Extended Experiments

### C.1   Experimental Setting

**Hyperparameters:** Our training procedure is divided into three main stages. The first stage is Thyme-SFT, which begins with training on image manipulation-related data, followed by annealing on mathematical code-related data. The learning rates for these two phases are set to $1 \times 10^{-5}$ and $1 \times 10^{-6}$, respectively, while all other hyperparameters remain consistent. The batch size is set to 128, and a total of 3 epochs are trained. The checkpoint from the final epoch of this stage is used to initialize the subsequent phase. The warmup ratio is configured to 0.05. The reinforcement learning stage utilizes a learning rate of $5 \times 10^{-7}$, with training conducted for 1 epoch. The KL divergence coefficient is set to 0.001, the batch size is 256, the number of rollouts is 4, and the repetition penalty is configured to 1.05. We select Qwen 2.5 VL 7B as the backbone (Bai et al., 2025a). All training is conducted on 32 NVIDIA H800 GPUs, requiring approximately 224 GPU hours in total. The annealing stage

Table 3: **User Prompt for SFT.**

---

```
<image>
```

**User's Question:** `[User Question]`

**User Image Path:** `[Image Path]`

**User Image Size:** `[Image Size]`

**Output Format (strict adherence required):**

```
<think>Your detailed reasoning process, including any code,
should go here.</think>
```

```
<answer>Your final answer to the user's question goes
here.</answer>
```

---

consumes around 8 GPU hours, whereas the reinforcement learning stage demands over 1200 GPU hours.

## C.2 BENCHMARKS AND BASELINES

**Benchmarks and metrics:** We mainly select three categories of benchmarks. The first category focuses on perception tasks because Thyme's image operations mainly aim to enhance perception ability. These benchmarks include the MME-RealWorld (Zhang et al., 2024a) series, HR Bench (Wang et al., 2025b), V* (Wu and Xie, 2024), RealWorld QA (xAI, 2024), etc. We report results for different splits of each benchmark. For example, for the MME-RealWorld series, we report perception and reasoning accuracy separately. For HR Bench, we report Fine-grained Single-instance Perception (FSP) and Fine-grained Cross-instance Perception (FCP) separately. For V* (Wu and Xie, 2024), we report recognition and spatial relationship reasoning performance. The second category is reasoning tasks since Thyme not only manipulates images but also can transform complex computations into code, thus its reasoning ability is also strengthened. Therefore, we select MathVision (Wang et al., 2024), MathVista (Lu et al., 2024), MathVerse (Zhang et al., 2024b), LogicVista (Xiao et al., 2024), WeMath (Qiao et al., 2024), and VisuLogic (Xu et al., 2025) to verify the model's reasoning ability. The third category consists of more general tasks, which primarily verify whether the model gains some performance improvement in general tasks as perception and reasoning abilities increase. These benchmarks include Hallucination bench (Li et al., 2023), MMStar (Chen et al., 2024), MMVet Hard (Yu et al., 2024), OCR Bench (Liu et al., 2023), Chart QA (Masry et al., 2022), and BLINK (Fu et al., 2024a).

**Baselines:** We take Qwen-2.5-VL-7B (Bai et al., 2025b) as the primary baseline. Additionally, we compare the performance of Thyme-7B with other MLLMs such as InternVL3-8B (Zhu et al., 2025), as well as larger-scale models including Qwen-2.5-VL-32B and the closed-source model GPT-4o (OpenAI, 2024). To ensure a fair comparison, we employ VLMEvalKit (Duan et al., 2024) as the evaluation pipeline.

## C.3 EVALUATION RESULTS

**Main Results:** Table 4 presents a comprehensive comparison between Thyme and other leading multimodal models across a range of benchmarks covering *Perception*, *Reasoning*, and *General* tasks. In perception tasks, Thyme demonstrates clear advantages even over larger-scale models such as Qwen2.5-VL 32B, indicating that simply scaling model size does not effectively address perception challenges. Instead, Thyme's test-time scaling strategy proves highly beneficial for perception tasks. Furthermore, by converting complex computations into executable code through training, Thyme achieves notable improvements in reasoning abilities. However, in this domain, the benefits of scaling model size are more pronounced,

Table 4: **Performance Comparison on Perception, Reasoning, and General Tasks.** For all open-source models, the best performance for each metric is **bolded**, and the second best is underlined. Gold-colored font indicates improvement over the baseline Qwen 2.5-VL 7B.

| Benchmark | Split | Thyme-VL 7B | Qwen2.5-VL 7B | InternVL3 8B | Qwen2.5-VL 32B | GPT-4o |
|---|---|---|---|---|---|---|
| *Perception* | | | | | | |
| HRbench-4K | FSP | **91.0**$_{+5.8}$ | 85.2 | 78.8 | 87.5 | 66.8 |
| | FCP | **63.0**$_{+10.8}$ | 52.2 | 61.3 | 59.3 | 63.3 |
| | Overall | **77.0**$_{+8.2}$ | 68.8 | 70.0 | 73.4 | 65.0 |
| HRbench-8K | FSP | **86.5**$_{+7.7}$ | 78.8 | 78.8 | 82.3 | 60.8 |
| | FCP | 57.5$_{+5.7}$ | 51.8 | **59.8** | 58.5 | 58.5 |
| | Overall | **72.0**$_{+6.7}$ | 65.3 | 69.3 | 70.4 | 59.6 |
| MME-Real | Perception | **67.1**$_{+6.5}$ | 60.6 | 63.5 | 63.8 | 64.9 |
| | Reasoning | **48.4**$_{+9.8}$ | 38.6 | 44.9 | 40.4 | 47.3 |
| | Overall | **64.8**$_{+6.6}$ | 58.3 | 61.3 | 61.0 | 62.8 |
| MME-Real-CN | Perception | **70.5**$_{+2.5}$ | 68.0 | 65.5 | 68.0 | 63.6 |
| | Reasoning | **52.1**$_{+6.5}$ | 45.6 | 50.0 | 44.4 | 51.3 |
| | Overall | **64.6**$_{+3.8}$ | 60.8 | 60.5 | 60.5 | 59.7 |
| MME-Real-Lite | Perception | **59.1**$_{+10.4}$ | 48.8 | 51.0 | 50.6 | 54.4 |
| | Reasoning | **49.1**$_{+11.3}$ | 37.7 | 44.8 | 39.3 | 48.3 |
| | Overall | **55.2**$_{+11.1}$ | 44.1 | 48.6 | 46.2 | 52.0 |
| V* | Attribute | **83.5**$_{+5.3}$ | 78.2 | 67.8 | 77.4 | 72.2 |
| | Spatial | 80.3$_{+6.7}$ | 73.6 | 73.7 | **86.8** | 60.5 |
| | Overall | **82.2**$_{+5.8}$ | 76.4 | 70.2 | 81.2 | 67.5 |
| RealWorld QA | Overall | **70.2**$_{+2.0}$ | 68.2 | 70.0 | 70.2 | 75.5 |
| *Reasoning* | | | | | | |
| MathVision | Mini | 27.6$_{+0.6}$ | 27.0 | 26.3 | **35.2** | 36.5 |
| MathVista | Mini | 70.0$_{+1.8}$ | 68.2 | 70.4 | **72.2** | 63.4 |
| MathVerse | Vision Only | 39.1$_{+3.9}$ | 35.2 | 29.2 | **40.0** | 35.3 |
| LogicVista | Overall | 49.0$_{+9.2}$ | 39.8 | 45.6 | **54.4** | 53.2 |
| WeMath | Overall | 39.3$_{+5.0}$ | 34.3 | 31.7 | **47.1** | 44.2 |
| VisuLogic | Overall | 23.4$_{+3.4}$ | 20.0 | 24.9 | **25.8** | 26.7 |
| *General* | | | | | | |
| Hallucination | aAcc | 71.0$_{+5.4}$ | 65.6 | 65.9 | **71.2** | 65.2 |
| | fAcc | 48.3$_{+9.4}$ | 38.8 | 41.3 | **50.6** | 44.8 |
| | qAcc | 47.7$_{+7.3}$ | 40.4 | 40.7 | **49.2** | 40.7 |
| | Overall | 55.6$_{+7.3}$ | 48.3 | 49.3 | **57.0** | 50.2 |
| MMStar | Overall | 65.9$_{+1.2}$ | 64.7 | 68.5 | **69.1** | 65.7 |
| MMVet Hard | Overall | **58.3**$_{+5.5}$ | 52.9 | 55.1 | 48.4 | 58.3 |
| OCR Bench | Overall | 86.3$_{-2.1}$ | **88.4** | 88.1 | 85.5 | 809.0 |
| Chart QA | Human | **80.0**$_{+7.5}$ | 72.5 | 77.0 | 76.9 | 79.5 |
| | Augment | 92.2$_{-2.7}$ | **94.9** | **94.9** | 82.6 | 91.9 |
| | Overall | **86.1**$_{+2.4}$ | 83.7 | 85.9 | 81.1 | 85.7 |
| BLINK | Val | 56.1$_{-0.3}$ | 56.4 | 55.5 | **63.6** | 63.3 |

suggesting that reasoning and logical inference capabilities largely depend on the inherent knowledge within the model itself. Thyme mainly enhances visual recognition quality and helps avoid the model independently predicting overly complex computations. Finally, due to the improvements in both perception and reasoning, Thyme shows significant gains in many general tasks, particularly with a substantial reduction in hallucination.

### C.4 EXPANDED BASELINES, EVALUATION SETUP, AND COMPARISON ANALYSIS

**Expanded Baselines.** To provide a more thorough comparison, we include a new set of strong baselines covering both commercial APIs and recent academic tool-use methods:

- **State-of-the-art commercial APIs:** GPT-4o, Gemini Flash 2.5, Gemini Pro 2.5, Claude 4 Sonnet, and OpenAI O3.
- **Recent academic works:** ZoomEye (Shen et al., 2025a), DyFo (Li et al., 2025a), Chain-of-Focus (Zhang et al., 2025c), Pixel Reasoner (Su et al., 2025), DeepEyes (Zheng et al., 2025), and Mini-O3 (Lai et al., 2025) and Skywork-R1V4 (Zhang et al., 2025f).

**Evaluation Setup (Representative Subset).** Due to time constraints, we report results on a representative subset of benchmarks. For the API models, we evaluate performance on **VStar-Bench**, **HRBench4K**, **MME-RealWorld**, **LogicVista**, **MathVision-MINI**, **MMVet-Hard**, and **MMStar**. This subset covers *three perception* benchmarks (VStarBench, HRBench4K, MME-RealWorld), *two reasoning* benchmarks (LogicVista, MathVision-MINI), and *two general-purpose* benchmarks (MMVet-Hard, MMStar). For recent methods (e.g., DeepEyes, Visual Sketchpad), we align evaluation on common benchmarks whenever possible for direct comparison.

**Conceptual Comparison with Prior Work.** We categorize prior tool-use approaches into three groups to clarify methodological differences and highlight the novelty of Thyme:

1. **Training-free methods.** These methods prompt an LLM to generate code or plans to orchestrate predefined visual APIs, without task-specific training. Representative examples include ViperGPT (Surís et al., 2023) and VisProg (Gupta and Kembhavi, 2023), which predefine visual APIs (e.g., detection, segmentation) and rely on code-based orchestration, and Visual Sketchpad (Hu et al., 2024), which allows an off-the-shelf MLLM to draw on a visual sketchpad (boxes/lines/masks) for subsequent reasoning. PyVision (Zhao et al., 2025a) prompts a model to generate PIL-based manipulation code. As they do not undergo task-specific training, performance can be sensitive to prompting quality and handcrafted rules.
2. **Code-based image manipulation methods.** These methods generate code to perform visual modifications. For example, Refocus (Fu et al., 2025) generates code to draw boxes, masks, or highlights to focus attention. However, the potential of code generation for richer operations (e.g., iterative magnification, rotation correction, contrast enhancement, and computation) is not fully exploited.
3. **Single external tool approaches.** Methods such as Pixel Reasoner (Su et al., 2025), DeepEyes (Zheng et al., 2025), Mini-O3 (Lai et al., 2025), and Visual CoT (Shao et al., 2024b) treat crop-and-zoom as a discrete external tool to enhance perception. Compared to Thyme, their toolsets are more limited, which restricts flexibility on diverse reasoning and computation-heavy tasks.

**Performance Comparison and Analysis.** We summarize the results in Table 5. First, we observe that state-of-the-art open-source models such as DeepEyes and Thyme can achieve perception performance that is broadly comparable to Gemini Flash 2.5 on several benchmarks. (We note that VStarBench includes a training set; the suboptimal performance of some closed-source models may be partly due to domain mismatch.) However, a gap remains relative to top-tier models such as Gemini Pro 2.5 and O3.

Second, compared to training-free methods (e.g., Visual Sketchpad and PyVision-GPT-4.1), Thyme shows a clear advantage, indicating the importance of end-to-end training for reliable

Table 5: **Expanded baselines and results on a representative subset of benchmarks.** Dashes indicate results not available or not evaluated under a unified setting.

| Method | VStarBench | HRBench4K | MME-RealWorld | LogicVista | MathVision-MINI | MMStar | MMVet-Hard |
|---|---|---|---|---|---|---|---|
| **Tool-Use Methods** | | | | | | | |
| Visual Sketchpad | 80.3 | - | - | - | - | - | - |
| PyVision-GPT-4.1 | 75.9 | - | - | - | - | - | - |
| ZoomEye | 90.6 | 69.6 | - | - | - | - | - |
| DyFo | 81.2 | - | - | - | - | - | - |
| Chain-of-Focus | 88.0 | - | - | - | - | - | - |
| Pixel Reasoner | 86.3 | 74.0 | 64.4 | - | - | - | - |
| DeepEyes | 83.3 | 73.2 | 64.0 | 47.7 | 26.6 | 63.87 | 53.23 |
| Mini-O3 | 88.2 | 77.5 | 65.5 | - | - | - | - |
| Skywork-R1V4 | 88.0 | 82.8 | 71.4 | - | - | - | - |
| **Our Methods** | | | | | | | |
| Qwen2.5-VL-7B (Direct) | 76.4 | 68.8 | 58.3 | 39.8 | 27.0 | 64.5 | 52.9 |
| Thyme-VL-7B | 82.2 | 77.0 | 64.8 | 49.0 | 27.6 | 65.9 | 58.3 |
| **Commercial APIs** | | | | | | | |
| GPT-4o | 67.5 | 65.0 | 62.8 | 53.2 | 36.5 | 65.7 | 58.3 |
| Gemini Flash 2.5 | 73.82 | 79.25 | 60.9 | 47.65 | 35.19 | 70.0 | 73.19 |
| Gemini Pro 2.5 | 79.05 | 83.87 | 71.27 | 46.3 | 33.55 | 72.0 | 76.97 |
| Claude 4 Sonnet | 56.4 | 49.37 | 59.87 | 63.53 | 48.0 | 54.8 | 55.45 |
| O3 | 95.7 | 84.82 | 72.48 | 67.33 | 63.81 | 75.6 | 67.24 |

tool/code usage. Third, compared to crop-and-zoom-only approaches (e.g., DeepEyes and Mini-O3), Thyme is competitive on perception and exhibits stronger gains on reasoning benchmarks, consistent with the benefits of a more versatile toolset (image manipulations + computation).

## C.5 ABLATION AND ANALYSIS

### C.5.1 THE IMPACT OF TRAINING STRATEGIES ON THE SFT PROCESS

In this subsection, we conduct detailed ablations on different strategies during the SFT phase. The specific results appear in Table 6. Below we introduce and compare these strategies.

⬦ **Naive SFT:** This approach directly uses all data for SFT without incorporating any of our training strategies. Direct SFT does not yield significant performance gains, and the model's output format is quite chaotic. During inference, it tends to predict the content of the sandbox on its own, and the code blocks and sandbox outputs are sometimes overwritten by the model's own predictions.

⬦ **Mask Sandbox:** In the SFT stage, we mask sandbox labels and outputs. This requires the model only to learn how to write code and how to perform further reasoning based on the returned results, leading to a significant enhancement in performance.

⬦ **Only Last Round:** This strategy masks information from previous rounds. For tasks involving multi-round code execution, the model only needs to learn the output of the final round. This effectively prevents the model from learning strange patterns, such as a tendency to produce a suboptimal piece of code in the first round and then correct it in subsequent rounds.

⬦ **Without Code Comment:** We also investigate the impact of the presence of comments in the code on the model's performance. In this line of experiments, we remove all comments from the code in the SFT data for training, but we find the results are not satisfactory. We speculate the reason is that while comments do not play a role during code execution, the process of writing comments implies an understanding of the code, making the model more logical when it writes code.

⬦ **Math Data Annealing:** Since our volume of code data for mathematical calculations is relatively small, when it is mixed with the full dataset for training, the model is rarely observed to write computation-related code during the inference phase. The primary goal of the SFT stage is to enable the model to learn this behavior. Therefore, although annealing this portion of the data does not show significant performance gains compared to the "Only Last Round" strategy, it more effectively introduces the behavior of using code to perform complex calculations.

Table 6: **Impact of Different SFT Strategies.** Directly mixing all SFT datasets for training does not yield optimal results. Instead, targeted strategies are necessary, such as masking sandbox content in the SFT data, training multi-turn interactive tasks using only the final turn, and employing annealing training on small batches of math calculation data to teach the model corresponding coding behaviors effectively.

| Training Data | Benchmark | Hallucination | MME-Realworld-Lite | | V* | HRBench 8K | MathVista | RealWorld QA |
| | Split | Overall | Perception | Reasoning | Overall | Overall | mini | Overall |
|---|---|---|---|---|---|---|---|---|
| - | Qwen 2.5 VL 7B | 48.29 | 48.75 | 37.73 | 76.40 | 65.50 | 68.20 | 68.20 |
| | Ours | | | | | | | |
| All | Naive SFT | 42.20 | 43.97 | 36.00 | 72.72 | 69.75 | 67.60 | 65.62 |
| | + Mask Sandbox | 50.20 | 50.10 | 42.90 | 78.50 | 65.34 | 68.70 | 69.01 |
| | + Only Last Round | 52.88 | 50.47 | 45.46 | 78.53 | 65.12 | 68.40 | 69.67 |
| | + wo Code Comment | 51.43 | 49.13 | 44.27 | 77.48 | 64.90 | 66.70 | 68.32 |
| Math Data | Annealing | 53.74 | 51.75 | 45.40 | 79.58 | 65.12 | 68.70 | 69.80 |

Table 7: **Investigation of Different Reward Design Strategies**. An ablation study is performed on various reward components, which are added to a baseline reward focused on outcome and format. The study examines the effects of incorporating a `Consistency` reward (evaluating alignment between the reasoning and the final answer), a `Process` reward (scoring the quality of the thinking process), and a `Code` reward (based on the successful execution of generated code). Observations show that adding a `Consistency` reward leads to consistent performance improvements. In contrast, the `Process` and `Code` rewards do not yield a positive gain, with the `Process` reward even having a negative impact on performance.

| Training Data | Benchmark | Hallucination | MME-Realworld-Lite | | V* | HRBench 8K | MathVista | RealWorld QA | Avg |
| | Split | Overall | Perception | Reasoning | Overall | Overall | mini | Overall | |
|---|---|---|---|---|---|---|---|---|---|
| | Baselines | | | | | | | | |
| - | Qwen 2.5 VL 7B | 48.29 | 48.75 | 37.73 | 76.40 | 65.50 | 68.20 | 68.20 | 59.01 |
| Thyme-SFT Data | Thyme-SFT | 53.74 | 51.75 | 45.40 | 79.58 | 65.12 | 68.70 | 69.80 | 62.01 |
| | Reward Design | | | | | | | | |
| Thyme-RL Data | Outcome+Format | 58.20 | 53.70 | 45.60 | 80.10 | 70.75 | 67.40 | 69.41 | 63.59 |
| | + Consistency | 56.66 | 58.25 | 49.06 | 81.76 | 72.25 | 69.70 | 72.06 | 65.68 |
| | + Process Reward | 55.63 | 52.95 | 44.60 | 80.10 | 72.25 | 67.50 | 67.32 | 62.91 |
| | + Code Reward | 52.18 | 56.80 | 49.86 | 82.72 | 70.87 | 69.10 | 70.28 | 64.54 |

### C.5.2 THE INFLUENCE OF RL REWARD DESIGN ON THE RL PROCESS

In this subsection, we explore a variety of reward design strategies, summarize the challenges encountered, and share insights with the aim of informing future work, as shown in Table 7. Besides format rewards and outcome rewards, the main additional reward functions include:

◇ **Consistency reward:** We provide Qwen2.5-VL-72B with the last 500 characters of the thinking process (usually a summary) and the answer content, asking it to assess the consistency between the reasoning and the final answer.

◇ **Code reward:** This reward is the proportion of model-generated code that runs successfully, calculated by dividing the number of successfully executed codes by the total number of codes written. This reward encourages the model to write more code and ensure its correctness.

◇ **Process reward:** The entire thinking process is fed into Qwen2.5-VL-72B, which assigns a score based on the text quality and reasoning logic.

All additional rewards apply only when the model's answer is correct. This prevents the model from biasing toward generating code or maintaining consistency at the expense of solving the problem correctly and providing accurate answers. In other words,

$$\text{Reward } r = \text{Result Reward} \times \left(1 + 0.5 \times \text{Additional Reward}\right) + 0.5 \times \text{Formatting Reward}.$$

As we can see from Table 7, the consistency reward brings consistent performance improvements, which aligns with observations from existing work. Scoring the quality of the reasoning process increases the computational overhead for the reward but, conversely, has a negative impact on performance. This is largely because the process score is highly subjective and easy to "hack". Similarly, during training, we observe that the code reward does, to

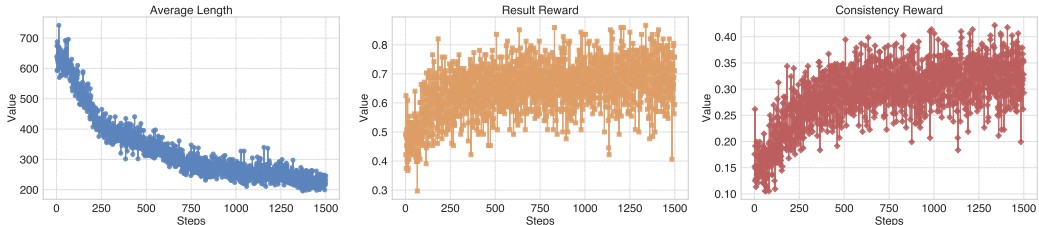

Figure 7: **Key Metrics During the RL Training Phase.** The plots illustrate the dynamic changes in average response length, accuracy reward, and consistency reward over the training steps.

some extent, encourage the model to write more code to help answer the question. However, this code is not always essential for the problem—for instance, cropping an uncomplicated graph, performing a very simple addition with code, or even generating a code block that contains only comments. Consequently, it does not yield a positive gain in performance.

### C.5.3 ANALYSIS OF THE RL LEARNING PROCESS

Figure 7 illustrates the dynamics of several critical metrics during the RL process, from which we derive the following observations and findings.

First, the average response length shows a rapid initial decrease before gradually converging. This phenomenon stems from the SFT phase, which introduces a large volume of synthetically generated image operation data to teach the model code generation for problem-solving. However, the images in this dataset are often not complex, rendering many of the coding operations non-essential. During the RL phase, the model quickly discerns that for a majority of tasks, a direct textual response is more efficient than a code-based analysis. Consequently, there is a swift reduction in the generation of unnecessary code. Second, the result reward exhibits a generally upward trend, which aligns with our expectations for a successful learning process. A more noteworthy observation pertains to the consistency reward. In the initial training stages, the average result reward is approximately 0.5, whereas the average consistency reward is only 0.15. This disparity indicates that a significant number of initially correct answers lack coherent reasoning, highlighting the critical role of the consistency reward. By incorporating this reward, the model is incentivized to align its reasoning with its final output. As training progresses, the result reward converges to approximately 0.7, while the consistency reward increases to nearly 0.35. This demonstrates that the inclusion of the consistency reward substantially mitigates the problem of contradictions between the model's reasoning and its answers.

### C.6 COMPARISON WITH CONCURRENT WORK AND DIRECT RL

We compare our model with the concurrent work Deepeyes-7B (Zheng et al., 2025), which only supports cropping functionality. Using identical evaluation configurations, the experimental results are shown in Table 8. Using identical evaluation configurations, our Thyme-VL-7B consistently outperforms all baselines across multiple benchmarks. On Hallucination Bench, our model achieves 71.00% aAcc compared to Deepeyes-7B's 59.31% and the base model's 65.61%. Similar improvements are observed on MME-Realworld-Lite, where we achieve 59.10% perception and 49.10% reasoning scores, and on HR8K (72.00%) and MathVista (70.00%). Additionally, we compare against directly applying RL to the base model, which shows only moderate improvements (68.24% vs 65.61% aAcc). Our method significantly outperforms this direct RL approach, validating the value of long chain-of-thought cold-start initialization.

Table 8: Comparison with concurrent work and direct RL approach across multiple benchmarks.

| Models | Hallucination Bench | | | MME-Realworld-Lite | | HR8K | MathVista |
|--------|------|------|------|------------|-----------|------|-----------|
| | aAcc | fAcc | qAcc | Perception | Reasoning | | |
| Qwen2.5-VL-7B | 65.61 | 38.83 | 40.43 | 48.75 | 37.73 | 65.50 | 68.20 |
| Direct RL | 68.24 | 44.95 | 44.05 | 52.65 | 43.66 | 67.35 | 69.24 |
| Deepeyes-7B | 59.31 | 34.10 | 33.19 | 53.97 | 46.00 | 69.75 | 67.60 |
| **Thyme-VL-7B** | **71.00** | **48.30** | **47.70** | **59.10** | **49.10** | **72.00** | **70.00** |

## C.7 CASE STUDIES

### C.7.1 CROPPING & ZOOMING

**Question:** What street is indicated by the sign on the pole near the center bottom of the image? Options: A. MACHIGAN B. MACHLGUN C. MICHIGUN D. MICHIGAN
Please select the correct answer from the options above.
**Model Responses:** `<think>`
The task involves identifying the street name indicated by the sign located near the center bottom of the image. The sign appears small and somewhat distant, making it challenging to read clearly without zooming in. Cropping and zooming in on this specific region will enhance the visibility of the text on the sign, allowing for accurate identification of the street name. The sign is positioned near the lower part of the image, slightly to the right of the center. By cropping this area, we can focus on the sign and improve its readability. The coordinates for cropping can be estimated based on the sign's position relative to the entire image. Here is the Python code to crop and zoom in on the relevant section:

```python
import os
import random
from PIL import Image

# Load the image
image_path = "hr_bench_4k/179.jpg"
image = Image.open(image_path)

# Define the cropping coordinates (x1, y1, x2,
y2)
# These coordinates are estimated based on the
sign's position in the image
x1, y1, x2, y2 = 1550, 2650, 1850, 2750

# Crop the image
cropped_image = image.crop((x1, y1, x2, y2))

# Apply a zoom factor (optional, depending on
the required level of detail)
zoom_factor = 2
zoomed_image =
cropped_image.resize((cropped_image.width *
zoom_factor, cropped_image.height *
zoom_factor))

# Save the processed image with a random suffix
random_suffix = random.randint(1000, 9999)
processed_filename = f"179_{random_suffix}.jpg"
processed_path =
"/mnt/data/temp_processed_images/" +
processed_filename
zoomed_image.save(processed_path)

# Print the path of the saved cropped image
print(processed_path)
```

<sandbox_output> <subimage_sign> </sandbox_output>
Upon examining the cropped and zoomed-in image, the sign clearly reads Ν MICHIGAN AV 100 E: This confirms that the street indicated by the sign is Michigan Avenue. `</think>`
`<answer>` D. MICHIGAN `</answer>`

Figure 8: **Cropping & Zooming Case 1.** Thyme first evaluates the size and distance of the sign, determining that cropping and zooming in on the corresponding area would improve visibility. It then proceeds to write code to crop and enlarge the region containing the sign. Thyme accurately locates the sign's position, successfully crops and zooms in on the area, and correctly answers the question.

**Question:** What is the phone number listed on the building in the foreground? Options: A. 206 441 5000 B. 206 451 5000 C. 206 444 5000 D. 206 441 4000

Please select the correct answer from the options above.

**Model Responses:** <think>

The task involves identifying the phone number listed on a building in the foreground of the image. The number appears to be written vertically on a sign attached to a tall building. To accurately read the digits, it is necessary to zoom in on this specific area of the image. The sign is located on the left side of the building, near the center of the lower part of the image. Cropping and zooming in on this region will enhance the clarity of the text, making it easier to discern the correct digits. The coordinates for cropping are estimated based on the position of the sign relative to the entire image. The cropped area should focus on the vertical text on the building, which is approximately in the middle-left section of the image. This targeted cropping will isolate the relevant part of the image, improving readability. Here is the Python code to crop and zoom into the specified area:

```
import cv2
import os
import random

# Load the image
image_path = "hr_bench_4k/224.jpg"
image = cv2.imread(image_path)

# Define the cropping coordinates (x1, y1, x2,
y2)
x1, y1, x2, y2 = 1750, 1950, 1850, 2600

# Crop the image
cropped_image = image[y1:y2, x1:x2]

# Apply zoom (resize the cropped image to
enhance details)
zoom_factor = 2
zoomed_image = cv2.resize(cropped_image, None,
fx=zoom_factor, fy=zoom_factor,
interpolation=cv2.INTER_CUBIC)

# Save the processed image with a random suffix
random_suffix = random.randint(1000, 9999)
processed_filename = f"224_{random_suffix}.jpg"
processed_path =
"/mnt/data/temp_processed_images/" +
processed_filename
cv2.imwrite(processed_path, zoomed_image)

# Print the path of the saved cropped image
print(processed_path)
```

<sandbox_output> <subimage_phone_number> </sandbox_output>
Upon examining the cropped and zoomed-in image, the phone number listed on the building is clearly visible. The digits are arranged vertically, and they read 206 441 5000. This matches option A. The other options do not correspond to the digits shown on the building's sign. </think>
<answer> A. 206 441 5000 </answer>

Figure 9: **Cropping & Zooming Case 2.** Thyme accurately locates the phone number's position, successfully crops and zooms in on the area, and correctly answers the question.

**Question:** What is the number of awning-tricycles in the image? Options: A. 78 B. 72 C. 43 D. 63 E. The image does not feature the awning-tricycles Please select the correct answer from the options above.

**Model Responses:** `<think>`

To determine the number of awning-tricycles in the image, we need to carefully examine the entire scene. The awning-tricycles are likely small objects that might be difficult to spot due to their size relative to the overall image. Cropping and zooming in on specific regions where awning-tricycles might be present can help identify them more accurately. The awning-tricycles appear to be located near the lower part of the image, close to the road and possibly parked or moving along the street. By cropping and zooming in on this area, we can focus on the relevant details and count the awning-tricycles more precisely. Let me proceed with cropping and zooming in on the lower part of the image where awning-tricycles are likely to be found:

```python
import os
import random
from PIL import Image

# Load the image
image_path = "MME-RealWorld/10993.jpg"
image = Image.open(image_path)

# Define the coordinates for cropping (based on
the inferred region)
x1, y1, x2, y2 = 0, 1000, 1000, 1500  # Adjust
these coordinates based on the image content

# Crop the image
cropped_image = image.crop((x1, y1, x2, y2))

# Apply zoom (resizing the cropped image for
better visibility)
zoom_factor = 2
zoomed_image =
cropped_image.resize((cropped_image.width *
zoom_factor, cropped_image.height *
zoom_factor))

# Save the processed image with a random suffix
random_suffix = random.randint(1000, 9999)
processed_filename =
f"10993_{random_suffix}.jpg"
processed_path =
"/mnt/data/temp_processed_images/" +
processed_filename
zoomed_image.save(processed_path)

# Print the path of the saved cropped image
print(processed_path)
```

`<sandbox_output>` `<subimage_tricycle>` `</sandbox_output>`
Upon analyzing the cropped and zoomed-in image, it becomes clear that there are no awning-tricycles visible in the scene. The focus of the image is on a road with vehicles and buildings, and no such objects can be identified. Therefore, option E is correct as the image does not feature awning-tricycles. `</think>`

`<answer>` E. The image does not feature the awning-tricycles `</answer>`

Figure 10: **Cropping & Zooming Case 3.** Thyme first analyzes the size of the awning-tricycles and the image, concluding that enlarging the area most likely to contain awning-tricycles could help determine the answer to the question. It then crops and zooms in on the lower-left corner of the image for closer inspection, ultimately concluding that there are no awning-tricycles in the picture.

C.7.2 ROTATION & CONTRAST ENHANCEMENT

**Question:** Please write out the expression of the formula in the image using LaTeX format.

**Model Responses:** <think> The text in the image appears to be slanted and difficult to read in its current orientation. To improve clarity, rotating the image by a specific angle will help align the text properly and make it easier to interpret. Let me proceed with the rotation to enhance readability.

```python
import cv2
import numpy as np
import os

# Load the image
image_path = "OCR_55251.png"
image = cv2.imread(image_path)

# Define the rotation angle (in degrees)
angle = 172  # The angle is derived from the
filename suffix

# Get the image dimensions
height, width = image.shape[:2]

# Calculate the rotation matrix
rotation_matrix = cv2.getRotationMatrix2D((width
/ 2, height / 2), angle, 1)

# Perform the rotation
rotated_image = cv2.warpAffine(image,
rotation_matrix, (width, height))

# Save the rotated image with a random suffix
random_suffix = np.random.randint(100000,
999999)
processed_path =
f"/mnt/data/temp_processed_images/OCR_55251_{random_suf
cv2.imwrite(processed_path, rotated_image)

# Print the saved image path
print(processed_path)
```

<sandbox_output> <subimage_OCR> </sandbox_output> After rotating the image, the text becomes more legible. The formula in the image is a decimal number with a bar over the last digit, which typically indicates a repeating decimal. The number is 0.642857, with the last digit, 7, being repeated indefinitely. This can be represented in LaTeX format as \0.\overline642857\. The bar over the digits 642857 signifies that these digits repeat continuously. This notation is commonly used to represent repeating decimals in mathematical expressions. </think>
<answer>

$$0.\overline{642857}$$

</answer>

Figure 11: **Rotation Case.** Thyme realizes that the input image is not properly oriented, so it performs a rotation operation with Python code to adjust the angle of the input image. Then, it correctly identifies the expression in the image and represents the result with LaTeX code.

**Question:** What is written in the image? Please try to answer the question with short words or phrases if possible.

**Model Responses:** `<think>`

The image appears to have low contrast, which may lead to poor text extraction in OCR. The text "communities" is somewhat faint and blends into the background, making it challenging for OCR systems to accurately recognize the characters. Enhancing the contrast will make the text stand out more clearly from the background, improving readability and OCR performance. To address this, we can use Python with the PIL library to enhance the contrast of the image. Here is the code to do so:

```python
import os
import random
from PIL import Image, ImageEnhance

# Load the image
image_path = "OCRBench/150.jpg"
img = Image.open(image_path)

# Enhance the contrast
enhancer = ImageEnhance.Contrast(img)
enhanced_img = enhancer.enhance(2.0)  # Increase
contrast by a factor of 2

# Save the enhanced image with a random suffix
random_suffix = random.randint(1000, 9999)
filename, ext =
os.path.splitext(os.path.basename(image_path))
processed_filename =
f"{filename}_{random_suffix}{ext}"
processed_path =
os.path.join("/mnt/data/temp_processed_images/",
processed_filename)
enhanced_img.save(processed_path)

# Print the path of the saved enhanced image
print(processed_path)
```

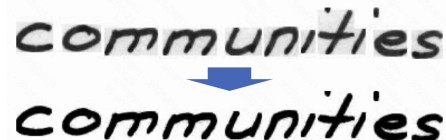

`<sandbox_output>` `<subimage_ocr>` `</sandbox_output>`
Upon examining the original image, the text "communities" is faint and blends into the background. After applying the contrast enhancement, the text becomes significantly more legible. The enhanced image clearly shows the word "communities," confirming that the enhancement process successfully improved the visibility of the text. `</think>`
`<answer>` communities `</answer>`

Figure 12: **Contrast Enhancement Case.** Thyme accurately locates the phone number's position, successfully crops and zooms in on the area, and correctly answers the question.

### C.7.3 COMPLEX CALCULATIONS

**Question:** A chef aboard a luxury liner wants to cook a goose. The time $t$ in hours to cook a goose at $180°$C depends on the mass of the goose $m$ in kilograms according to the formula

$$t = am^b$$

where $a$ and $b$ are constants. The table below gives the times observed to cook a goose at $180°$C.

| Mass, $m$ (kg) | Time, $t$ (h) |
|---|---|
| 3.00 | 2.75 |
| 6.00 | 3.75 |

Using the data in the table, determine both $a$ and $b$ to two decimal places.

**Model Responses:** `<think>`

To find the constants $a$ and $b$ in the equation $t = am^b$, we need to use the given data points from the table. The data points are: 1. $m_1 = 3.00$ kg, $t_1 = 2.75$ h 2. $m_2 = 6.00$ kg, $t_2 = 3.75$ h

We can set up two equations based on these data points: 1. $2.75 = a(3.00)^b$ 2. $3.75 = a(6.00)^b$

We can solve this system of equations for $a$ and $b$. Let's denote $x = a$ and $y = b$. Then the equations become: 1. $2.75 = x(3.00)^y$ 2. $3.75 = x(6.00)^y$

Dividing the second equation by the first equation eliminates $x$:

$$\frac{3.75}{2.75} = \frac{x(6.00)^y}{x(3.00)^y} = \left(\frac{6.00}{3.00}\right)^y = 2^y$$

So,

$$2^y = \frac{3.75}{2.75} = \frac{15}{11}$$

Taking the logarithm base 2 of both sides:

$$y = \log_2\left(\frac{15}{11}\right)$$

Using Python to compute this value:

```
import math

# Calculate y = log2(15/11)
y_val = math.log2(15/11)
print(y_val)
```

`<sandbox_output>` 0.44745897697122117 `</sandbox_output>`

So, $y \approx 0.45$. Now substitute $y = 0.45$ back into one of the original equations to find $x$. Using the first equation:

$$2.75 = x(3.00)^{0.45}; x = \frac{2.75}{3.00^{0.45}}$$

Using Python to compute this value:

```
import math

# Calculate x = 2.75 / (3.00**0.45)
x_val = 2.75 / (3.00**0.45)
print(x_val)
```

`<sandbox_output>` 1.6773671336980667 `</sandbox_output>`

So, $x \approx 1.68$.

Therefore, the constants are $a \approx 1.68$ and $b \approx 0.45$. `</think>`

`<answer>` 1.68, 0.45 `</answer>`

Figure 13: **Complex Calculations Case.** After deducing the mathematical expressions for x and y, Thyme writes code to perform the calculations, obtaining the accurate results.

## C.8 Bad Cases

### C.8.1 Complex Problems Without Coding

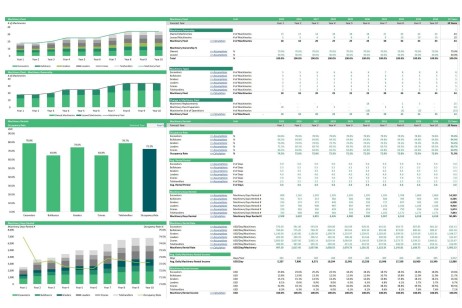

**Question:** Which machinery has the highest occupancy rate, according to the 'Occupancy Rate' diagram? Options: A. Exavators B. Graders C. Cranes D. Telehandlers E. The image does not feature the number. Please select the correct answer from the options above.

**Model Responses:** `<think>` The task is to identify which machinery has the highest occupancy rate based on the 'Occupancy Rate' diagram. The relevant section of the image is the bar chart labeled "Machinery Rentals - Occupancy Rate."This chart displays the occupancy rates for different types of machinery, with percentages shown for each type. The bar chart clearly shows that Telehandlers have the highest occupancy rate at 72.2%, followed by Excavators at 78.8%, Bulldozers at 64.8%, Graders at 74.0%, and Cranes at 64.8%. Therefore, option D, Telehandlers, is the correct choice. `</think>`

`<answer>` D. Telehandlers `</answer>`

Figure 14: **Failure Case 1.** In this case, the image resolution is high, but Thyme does not perform the necessary crop to focus on the most relevant area and instead answers directly.

### C.8.2 Unuseful Coding

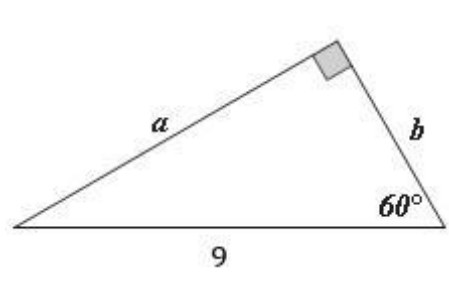

**Question:** Please directly answer the question and provide the final value, e.g., 1, 2.5, 300. Question: Solve for the unknown a. Please try to answer the question with short words or phrases if possible.

**Model Responses:** `<think>` To solve for the unknown side 'a' in the given right triangle, we can use trigonometric relationships. The triangle has a hypotenuse of length 9, one leg 'b' opposite the 60° angle, and the other leg 'a' adjacent to the 60° angle. We need to find the length of 'a'. Using the cosine function, which relates the adjacent side to the hypotenuse in a right triangle: cos(60°) = a / hypotenuse We know that cos(60°) = 0.5. So, substituting the known values: 0.5 = a / 9 To find 'a', we multiply both sides of the equation by 9: a = 0.5 * 9 Now, let's calculate the value of 'a':

```
# Calculate the value of 'a'
a_value = 0.5 * 9
print(a_value)
```

`<sandbox_output>` 4.5 `</sandbox_output>`
Therefore, the length of side 'a' is 4.5. `</think>`
`<answer>` 4.5 `</answer>`

Figure 15: **Failure Case 2.** The case requires only trivial calculations, yet Thyme opts to write code, adding unnecessary overhead. Furthermore, Thyme confuses variable $a$ with variable $b$, mistaking the value of $b$ for the value of $a$, which leads to an incorrect result.

### C.8.3 Inaccurate Cropping

**Question:** What is the model number of the train shown in the image? Options: A. 4005 B. 1002 C. 3001 D. 2001 Please select the correct answer from the options above.

**Model Responses:** `<think>`

The task is to identify the model number of the train shown in the image. The model number appears to be visible on the front of the train, near the bottom right corner of the train's front panel. The number is clearly visible as 2̈001.. To ensure accuracy and confirm this observation, it would be beneficial to zoom in on this specific area of the image. Cropping the image around the train's front panel will help focus on the model number and eliminate any distractions from the surrounding environment. The coordinates for cropping can be estimated based on the position of the train in the image. The relevant area is located towards the lower right part of the train's front panel. By cropping this section, we can zoom in on the model number for a clearer view. Here is the Python code to crop and zoom in on the relevant area:

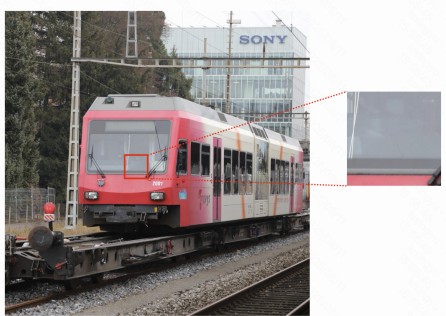

```python
import os
import random
from PIL import Image

# Load the image
image_path = "hr_bench_4k/243.jpg"
img = Image.open(image_path)

# Define the cropping coordinates (x1, y1, x2, y2)
# These coordinates are estimated based on the
position of the train's front panel
x1, y1, x2, y2 = 1600, 1900, 1900, 2200

# Crop the image
cropped_img = img.crop((x1, y1, x2, y2))

# Save the cropped image with a random suffix
random_suffix = random.randint(1000, 9999)
filename, ext =
os.path.splitext(os.path.basename(image_path))
processed_filename =
f"{filename}_{random_suffix}{ext}"
processed_path =
os.path.join("/mnt/data/temp_processed_images/",
processed_filename)
cropped_img.save(processed_path)

# Print the path of the saved cropped image
print(processed_path)
```

`<sandbox_output>` `<subimage_phone_number>` `</sandbox_output>`
The cropped image confirms that the model number of the train is indeed 2̈001.. This matches option D from the given choices. `</think>`
`<answer>` D. 2001 `</answer>`

Figure 16: **Failure Case 3.** In this case, although Thyme eventually arrives at the correct answer, the region it crops during thinking process is irrelevant to the problem.

# D ANNOTATION REQUIREMENTS

This annotation document aims to ensure each image accurately describes the characteristics of target objects during the annotation process and provides clear annotation information to users. The detailed steps and requirements are as follows:

## D.1 TASK DESCRIPTION

For each high-resolution image, annotators are required to complete the following tasks:

- **Question Annotation**: Design a specific question for each image. The target of the question should be small and difficult-to-recognize objects in the image, with the object occupying no more than 5% of the image resolution. Users need to zoom in on the image to clearly identify these small objects.

- **Answer Annotation**: Provide an accurate answer based on the question. The answer should be directly related to the question and derived from analyzing specific regions within the image.

- **Bounding Box**: Draw a bounding box for each annotated target object. The bounding box coordinates should be $(x_1, y_1, x_2, y_2)$, indicating the top-left and bottom-right corners of the object area. The bounding box should ensure that the cropped region clearly shows the target object.

- **Category**: Clearly specify the category of each question, selecting only from the five categories defined in the question design.

## D.2 ANNOTATION REQUIREMENTS

**Object Selection:** Select small objects in the image as targets. The area of these small objects should not exceed 5% of the total image area. Users need to zoom in to clearly identify these objects.

**Bounding Box Design:** The accuracy of bounding box coordinates need not be pixel-level but must ensure that the cropped area clearly displays the object. Please roughly determine the bounding box position and size according to the object features in the image.

**Question Design:** Questions should cover multiple aspects, including but not limited to:

- OCR recognition: Identify the text content in a certain location of the image. For example, "What is written on the sign in the image?"

- Attribute recognition: Identify attributes of specific objects in the image, such as color or shape. For example, "What is the color of this object?"

- Location recognition: Determine the location of objects within the image. For example, "Where is this object located in the image?"

- Quantity recognition: Identify the number of objects of the same type in the image. For example, "How many apples are in the image?"

- Object recognition: Identify the type of object at a specific location in the image.

- Chart understanding: For chart-type images, identify specific data points' values, compute maximum or minimum values, or predict trends. For example, "What is the maximum value in the chart?"

## D.3 ANNOTATION STEPS

1. **Image Review**: Open the high-resolution image and carefully examine all elements, especially small and hard-to-recognize objects.

2. **Object Localization**: Identify objects to be annotated, ensuring the object occupies no more than 5% of the image area. These are usually small objects or details in the background.

3. **Draw Bounding Box**: Use annotation tools to draw bounding boxes according to the shape and position of the target objects, ensuring the boxes fully contain the objects.

4. **Design Questions and Answers**: Based on specific image content, propose questions and provide appropriate answers. Questions should be concise and directly related to the target objects.

5. **Save Annotation Data**: Record annotation information (questions, answers, bounding box coordinates) for each image, ensuring all annotations are independent and meaningful.

## E  RELATED WORK

**Multimodal Large Language Models.**  Fueled by the advancements in LLMs the field of MLLMs has seen rapid development in recent years, with model capabilities evolving at a remarkable pace (Fu et al., 2024b; Zhang et al., 2024a; Yu et al., 2025; Li et al., 2022). Modern MLLMs, such as Qwen2.5-VL (Bai et al., 2025a), GPT-4o (OpenAI, 2024), and LLaVA (Li et al., 2024a), have demonstrated impressive performance in processing high-resolution images and engaging in complex human-like dialogue. Research has diversified into numerous sub-domains, including extending context length (Shen et al., 2025b), improving computational efficiency (Zhang et al., 2025g), mitigating hallucinations (Lu et al., 2025), enhancing conversational abilities (Xiong et al., 2024), and achieving better alignment with human preferences (Zhang et al., 2025d). Concurrently, more sophisticated architectures have emerged. Omni-MLLMs are capable of processing a mix of modalities like speech, video, and images simultaneously (Li et al., 2025b; Zhao et al., 2025b; Fu et al., 2024c), while Unify-MLLMs can produce interleaved, mixed-modal outputs, such as generating an image with auxiliary lines to aid in solving a math problem (Xie et al., 2024; Team, 2024). These works showcase a clear trajectory towards more integrated and versatile multimodal interaction.

**Multimodal Reasoning.**  Enhancing the reasoning capabilities of MLLMs is a critical frontier. Recently, RL has become a prominent technique in the post-training of MLLMs, leading to significant gains in vision tasks (Liu et al., 2025; Shen et al., 2025c), multimodal reasoning (Huang et al., 2025; Peng et al., 2025; Meng et al., 2025), and even reward modeling itself (Zhang et al., 2025e). Compared to traditional methods like SFT or Direct Preference Optimization (DPO) (Rafailov et al., 2023), RL-based approaches have shown superior generalization and an ability to induce more complex, long-term reasoning capabilities, as demonstrated by models like DeepSeek-R1 (DeepSeek-AI, 2025). However, a significant limitation of many existing efforts is that they primarily focus on enhancing the textual reasoning chain. The visual input often serves as a static condition rather than an active component within the reasoning process. While some paradigms have emerged to "think with images", they are often limited to a single function like cropping (Zheng et al., 2025) or generating an auxiliary image (Chern et al., 2025). Our work, **Thyme**, directly addresses this gap by empowering the model to autonomously generate and execute code for a diverse range of image manipulations and computations. This allows the model to treat the image not just as input, but as a dynamic entity that can be actively interrogated and transformed as an integral part of its reasoning process.

Table 9: **Prompt template for training data generation.**

---

You are an advanced AI assistant tasked with generating training data for a complex image processing and question-answering task. Your role is to generate an ideal response containing a detailed thought process and specific executable Python code based on the user's question and the assumed condition of the image.

**User Input:** `<image>`

**User's Question:** `user question`

**User Image Path (just for code reference):** `user image path`

**Core Instructions:**

Your primary task is to determine whether the image can be used directly to answer the user's question or if it requires processing.

**1. If the image can be used directly:**

- Clearly state that the image is ready to be used without any processing.

- Do not generate any code in this case, just answer the question and include `-1` in the `<answer></answer>` box.

**2. If the image needs processing:**

a). Provide a detailed description of the issues with the image that prevent answering the question, such as incorrect orientation, low contrast, lighting issues, etc.

b). Choose the appropriate category of operation that addresses the identified issues:

  - **1. Direction Issues**: If the image needs rotation or flipping.

  - **2. Lighting and Contrast Issues**: If the image's brightness, contrast, or lighting needs adjustment, or the contrast between the text and background in the image is low for OCR.

  - **3. Scaling and Region of Interest (ROI)**: If parts of the image need to be cropped or resized.

  - **4. Combined Issues**: If more than one category applies, specify the primary issue category and reflect multi-step processing in the code.

c). Generate specific, executable Python code to address the identified image issues.

  - For example:

    - For **Direction Issues**: Specific angles for rotation (e.g., angle = 90 or angle = -90).

    - For **Scaling/ROI**: Specific coordinates like (x1, y1, x2, y2) for cropping.

  - Save the processed image in the temporary folder (`/mnt/data/temp_processed_images/`), with the same filename as the User Image Path, followed by a random suffix.

  - Print the saved image path (processed_path) in the last line to allow for further processing in a sandbox environment.

d) The code snippet must be wrapped with: ``

 ```python
 *code snippet*
 ```

 ``, and should be executable.

e). Output the Tool ID used, if applicable. If you used a tool, return the corresponding tool ID. For instance, if you used ROI-related code, return `<answer>3</answer>`.

**Output Format (strictly follow):**

```
<think>Your detailed comparative analysis and executable code goes
here</think><answer>Tool ID if you use tool else -1</answer>
```

---

Table 10: **Prompt template for visual QA with cropping based on bounding box.**

---

You are an advanced AI assistant tasked with constructing reasoning and code for a visual QA task. You will receive the user's question, image, User Image Path, and Ground Truth Bounding Box Coordinates. The image path and Ground Truth Bounding Box coordinates should only be used in the code and must not be mentioned in your analysis.

**User Input:** `<image>`

**User's Question:** `user question`

**User Image Path** (**just for code reference**)**:** `user image path`

**Ground Truth Bounding Box Coordinates:** `(x1, y1, x2, y2)`

**Core Instructions:**

Your primary task is to provide a reasonable explanation of why cropping the image is necessary to answer the user's question based on the provided bounding box coordinates.

**Important:** Do **not** mention the provided "Ground Truth Bounding Box" in your analysis. Treat the bounding box coordinates as something you have inferred based on the content of the image. You should only reference these coordinates in your **executable code**, and avoid explicitly stating them in your analysis or thought process.

a). Provide a detailed description of why cropping is necessary. For example:

- "The task seems to be extracting text from a sign near a door under a balcony. I guess I'll need to zoom in and crop the region around the sign. The coordinates appear to be near the center of the lower part of the image above the door. I'll refine this area further for better readability. Let me get started on that!"

b). Generate simple, executable Python code to crop the image based on the inferred bounding box coordinates.

- Save the processed image in the temporary folder (`/mnt/data/temp_processed_images/`), with the same filename as the User Image Path, followed by a random suffix.

- Print the saved image path (processed_path) in the last line to allow for further processing in a sandbox environment.

c) The code snippet must be wrapped with:

``

"'`python`

*code snippet*

`'` "

``, and should be executable.

**Output Format** (**strictly follow**)**:**

`<think>Your detailed analysis of why cropping is necessary and the executable code goes here.<think>`

`<answer>1</answer>`

---

Table 11: **Prompt template for visual QA with image rotation.**

---

You are an advanced AI assistant tasked with constructing reasoning and code for a visual QA task. You will receive the user's question, a rotated image, the rotation angle, and the user Image Path. The image path and rotation angle should only be used in the code and must not be mentioned in your analysis.

—

**User Input:** `<image>`

**User's Question:** `user question`

**User Image Path** (**just for code reference**): `user image path`

**GT Degree: How many degrees was the image rotated?**: `rotation angle`

**Core Instructions:**

Your primary task is to provide a reasonable explanation of why rotating the image is necessary to answer the user's question based on the provided rotation angle.

a). Provide a detailed description of why rotating the image is necessary. For example:

  - "The text in the image appears to be slanted and difficult to read in its current orientation. I believe rotating the image by a specific angle will help align the text properly and make it easier to interpret. Let me proceed with the rotation to improve clarity."

b). Generate simple, executable Python code to rotate the image by the inferred angle.

  - Save the processed image in the temporary folder (`/mnt/data/temp_processed_images/`), with the same filename as the User Image Path, followed by a random suffix.

  - Print the saved image path (processed_path) in the last line to allow for further processing in a sandbox environment.

c) The code snippet must be wrapped with:

  ``

  "'`python`

  *code snippet*

  "'

  ``, and should be executable.

The sum of your rotation angle and the GT Degree must be either 0 or 360; it should never be 180 or -180. And you do not need to answer the question.

**Important:** Do **not** mention the provided "GT Degree" in your analysis. Treat the GT Degree as something you've inferred based on the content of the image. You should only reference this angle in your **executable code**, and avoid explicitly stating it in your analysis, thought process, and comments in the code.

—

**Output Format (strictly follow):**

```
<think>Your detailed analysis of why rotating the image is
necessary and the executable code goes here.<think>
```

```
<answer>1</answer>
```

