# OpenReview forum: "Thyme: Think Beyond Images"
_ICLR.cc/2026/Conference — ICLR 2026 Poster_

### Official Review · Reviewer_7BVm · 2025-10-25

**Soundness:** 4
**Presentation:** 4
**Contribution:** 4
**Rating:** 10
**Confidence:** 5

**Summary:**

This paper proposes Thyme (Think Beyond Images), a framework that enables models to answer queries through a sequence of autonomous image operations such as cropping, rotating, and other manipulations. By combining supervised fine-tuning (SFT) for cold-start and reinforcement learning (RL) enhancement, Thyme achieves rich functionality via executable code generation. The results are impressive across several challenging benchmarks, particularly on high-resolution datasets like MME-RealWorld.

**Strengths:**

Overall, I enjoyed reading this paper and appreciate all the details. The followings are the parts that I found insightful

- Clear and well-structured presentation.
- Addresses an important problem area: reasoning in vision-language models (VLMs).
- Provides a detailed training recipe, including adaptive temperature, early stopping for repetitive n-grams, consistency rewards, math annealing, and loss applied on the last round of dialogue.
- Contains extensive implementation details in the Appendix, such as sandbox construction and other design choices.
- Strong ablation study (Table 5) demonstrating the effectiveness of different training details.
- Thorough evaluation of reward design choices in RL training (Table 6).

**Weaknesses:**

It would be valuable to see this approach extended along the temporal axis or even multi-image scenarios. Overall no major weakness in the paper.

**Questions:**

- What are the motivations for including mathematical computation data (as shown in Figure 4, right)? Specifically, these seem to involve pure math operations in code that are unrelated to visual operations—what purpose do they serve?
- What does the data distribution of the SFT dataset look like across the three difficulty categories?
- In Supplementary Section B, is it indeed describing the details of generating the SFT dataset?
- Approximately how many tokens are generated during the SFT phase and RL training phase?

---

> ### Author Response · Authors · 2025-11-16
> **Author Response**
>
> ### [Concern 1] What are the motivations for including mathematical computation data?
>
> Our goal is to train **Thyme not merely as an image manipulator**, but as a **general-purpose multimodal reasoning agent** capable of invoking computational tools—such as code for numerical or symbolic reasoning—whenever the task demands it. We believe coding capability is essential for future agents to generalize to unseen tool-based tasks (e.g., implementing novel MCP functions by code). In future work, we plan to expand beyond image operations and math to include other tool-augmented reasoning domains.
>
> ----
>
> ### [Concern 2] What does the data distribution of the SFT dataset look like across the three difficulty categories?
>
> The **Thyme-SFT dataset** (340K samples) is structured as follows:
> - **Category 1 (Direct Answer, no code)**: ~26% (87K samples)
> - **Category 2 (Single-turn code: image ops or math)**: ~29% (100K samples)
> - **Category 3 (Multi-turn interactive reasoning)**: ~45% (152K samples)
>
> This distribution reflects our design philosophy: maintain strong zero-shot reasoning (Cat. 1), ensure robust tool invocation (Cat. 2), and develop iterative, self-correcting behavior (Cat. 3).
>
> ----
>
> ### [Concern 3] In Supplementary Section B, is it indeed describing the details of generating the SFT dataset?
>
> Yes. **Appendix B (“Long-CoT Data Construction”)** provides:
> - The sourcing strategy for each sub-dataset (e.g., LLaVA-OV, V*),
> - Prompt templates tailored to each operation type (e.g., cropping, rotation, math computation).
> We will revise the appendix title for clarity (e.g., “SFT Data Construction”) and add explicit cross-references in the main paper to improve navigability.
>
> ----
>
> ### [Concern 4] Approximately how many tokens are generated during the SFT and RL phases?
>
> Thank you for this precise question. We provide token counts based on the **Qwen2.5-VL tokenizer**, applied to model *responses only* (i.e., output tokens).
>
> **SFT Phase:**
> We can accurately estimate token usage from the static Thyme-SFT dataset (340K samples):
> - **Category 1** (Direct Answer, no code): 87K samples → **19.96M tokens** (avg. 228.27 per response)
> - **Category 2** (Single-turn code: image ops or math): 100K samples → **51.12M tokens** (avg. 544.78)
> - **Category 3** (Multi-turn interaction): 152K samples → **104.78M tokens** (avg. 687.71)
> - **Additional pure math data** (12K samples): → **10.36M tokens** (avg. 1000.11)
>
> **Total SFT response tokens: ~186.2 million**
>
> **RL Phase:**
> Token count is dynamic due to policy improvement during training. The average response length decreases from ~700 to ~250–300 tokens over time. To estimate total consumption:
> - Training uses **55,200 unique prompts**
> - Each prompt generates **8 rollouts per iteration**
> - Assuming an **average of ~400 response tokens per rollout**,
>
> **Total RL response tokens ≈ 55,200 × 8 × 400 = 176.6 million**
>
> Thus, the entire training pipeline consumes approximately **362.8 million output tokens** (186.2M SFT + 176.6M RL). We note that this excludes input tokens and image tokens and focuses only on generated responses, which dominate the compute cost in our setting.

---

### Official Review · Reviewer_pa3i · 2025-10-28

**Soundness:** 3
**Presentation:** 3
**Contribution:** 3
**Rating:** 4
**Confidence:** 4

**Summary:**

This paper presents Thyme, an framework enabling multimodal large language models to autonomously generate and execute code for diverse image manipulations and mathematical computations. Building upon Qwen2.5-VL-7B, the work employs a two-stage training strategy: supervised fine-tuning on 500K curated samples followed by reinforcement learning with a novel GRPO-ATS algorithm featuring adaptive temperature sampling. The framework demonstrates impressive empirical results across ~20 benchmarks, achieving substantial improvements on challenging perception tasks and reasoning benchmarks. The engineering effort is commendable, including careful data construction, sandbox environment design, and comprehensive evaluation.

**Strengths:**

1. Comprehensive and well-engineered system: The paper presents a complete pipeline integrating multimodal understanding, code generation, and sandbox execution, with careful attention to practical details like error handling and security constraints.
2.  Rich functionality beyond existing work: Thyme supports diverse image manipulations (cropping, rotation, contrast enhancement) and mathematical computations, offering genuine multimodal reasoning capabilities.
3.  Well-designed data pipeline with multiple quality control stages and diverse task coverage including multi-turn interactions.
4.  GRPO-ATS with adaptive temperature sampling is a solution to the code generation reliability problem

**Weaknesses:**

1. The provided ablation studies (Table 5, 6, 7) suggest that the performance gains from several of these carefully designed components are limited.

2. The paper does not mention that the curated dataset and the specialized sandbox environment which are claimed as key components will be open-sourced. Whether these contents are open source is an important basis for judging the contribution of this paper.

**Questions:**

1. GRPO-ATS is one of the core contributions of this paper, but no ablation study is provided to verify its effectiveness. Although the temperature setting is clearly specified and intuitive motivation is provided, how to quantitatively analyze the effectiveness of this method is still unclear.

2. The curated dataset and the specialized sandbox environment will be open-sourced?

3. The paper presents "Naive SFT" in Table 5 as a baseline,  does this baseline have access to the meticulously annotated data that were exclusively prepared for RL?

---

> ### Author Response · Authors · 2025-11-16
> **Response [1/2]**
>
> ### [Concern 1] The provided ablation studies (Table 5, 6, 7) suggest that the performance gains from several of these carefully designed components are limited.
>
> We sincerely thank the reviewer for this thoughtful comment. To better address the concern, we would like to clarify which specific rows in Tables 5, 6, and 7 gave the impression that our proposed components yield limited gains—this may stem from our presentation not being sufficiently clear.
>
> We emphasize that **not all entries in these ablation tables represent core components of our final method**. Some are intentionally included as *negative results* or exploratory variants to inform the community about design choices that do **not** work well. Below, we provide further context for each table:
>
> - **Table 5**:
>   - *“– w/o Code Comment”* removes code comments during training, which leads to a performance drop—confirming that code comments are **retained** in our final model.
>   - *“Mask Sandbox”* and *“Only Last Round”* both provide consistent (though modest) improvements. While the gain from “Only Last Round” is small, it helps prevent undesirable behaviors—e.g., the model proposing an incorrect action in the first turn and “correcting” it in the second, which harms reasoning coherence.
>   - *“Math Data Annealing”* uses only a small batch of math-oriented data with the goal of **teaching the model to use code for numerical reasoning**, not to boost overall scores. The limited gain is expected; the key outcome is that the model learns to **invoke code for complex computation during reasoning**, which is a capability baseline models lack.
>
> - **Table 6**:
>   Both *“Process Reward”* and *“Code Reward”* were exploratory reward designs. We found that these more complex rewards **do not lead to significant performance improvements** over our simpler combination of *Outcome + Format + Consistency* rewards. Hence, they are **not used** in our final pipeline—this ablation serves to justify our reward design choice.
>
> - **Table 7**:
>   This table compares **Thyme-VL** (our full method) against **direct RL fine-tuning on the baseline model**. The performance gain is in fact **substantial**, demonstrating the effectiveness of our overall framework.
>
> If the reviewer still has concerns about specific rows or trends in these tables, we would greatly appreciate more detailed feedback so we can clarify further in the revision.
>
> ----
>
> ### [Concern 2] Will the dataset or the sandbox be released?
>
>
> We thank the reviewer for this important question. Please rest assured that **all of our data, training algorithms, and sandbox environment will be fully open-sourced** to support reproducibility and future research.
>
> Due to the large size of the supervised fine-tuning (SFT) dataset and time constraints during the review period, we have **already released** the following materials via an anonymous Hugging Face repository:
> 🔗 [https://huggingface.co/datasets/Openreview123/Thyme-Data](https://huggingface.co/datasets/Openreview123/Thyme-Data)
> This includes:
> - The reinforcement learning (RL) training data,
> - The complete sandbox environment used for code execution and validation,
> - All prompts and filtering scripts used in data curation.
>
> In the paper, we also provide a detailed description of our data generation pipeline and **publicly share all functional prompts** used to construct the dataset. We are committed to full transparency and will release the full SFT dataset and codebase upon publication.

---

> ### Author Response · Authors · 2025-11-16
> **Response [2/2]**
>
> ### [Concern 3] Ablation studies of GRPO-ATS
>
> Thank you for pointing this out. In response to the reviewer’s concern, we conducted an additional ablation using the exact same setup as in our original ablation studies. As shown in the last row of the table below (labeled “wo ATS”), **removing the Adaptive Temperature Sampling (ATS) module leads to a noticeable drop in performance**.
>
> Without ATS, the model quickly reduces its use of code generation during RL training. This is because stochastic sampling at higher temperatures produces code with frequent syntax or logic errors, which results in execution failures and zero reward. Consequently, the policy learns to avoid code altogether—hurting its ability to solve tasks that require programmatic reasoning. Although the performance drop appears modest in aggregate scores, it reflects a **fundamental degradation in the model’s tool-use behavior**, which ATS effectively mitigates.
>
> | Benchmark      | Hallucination | MME-Realworld-Lite |           | V*      | HRBench 8K | MathVista | RealWorld QA | Avg   |
> |----------------|---------------|--------------------|-----------|---------|------------|-----------|--------------|-------|
> | Split          | Overall       | Perception         | Reasoning | Overall | Overall    | mini      | Overall      |       |
> | Qwen 2.5 VL 7B |         48.29 |              48.75 |     37.73 |   76.40 |      65.50 |     68.20 |        68.20 | 59.01 |
> | Thyme-RL       |         56.66 |              58.25 |     49.06 |   81.76 |      72.25 |     69.70 |        72.06 | 65.68 |
> | wo ATS         |         54.60 |              53.62 |     45.20 |   78.72 |      70.12 |     68.20 |        70.71 | 63.31 |
>
> This result underscores that ATS is not merely an implementation detail but a **practical necessity** for stable and effective code-augmented reinforcement learning.
>
> ----
>
> ### [Concern 4]  The paper presents "Naive SFT" in Table 5 as a baseline, does this baseline have access to the meticulously annotated data that were exclusively prepared for RL?
>
> Thank you for the question. As noted in the caption of Table 5, the “Naive SFT” baseline refers to a model trained by mixing all available supervised fine-tuning (SFT) data—but it does not include any of the RL-specific data that was meticulously collected and filtered for reinforcement learning.

---

> > ### Author Response · Authors · 2025-11-26
> > **Kindly Request for Reviewer's Feedback**
> >
> > Dear Reviewer,
> >
> > Thank you so much for your time in improving our paper!
> >
> > Since the end of the rebuttal is coming soon, may we know if our response addresses your main concerns? Should you have any further advice, please let us know and we will be more than happy to engage in more discussion and improvements.

---

### Official Review · Reviewer_ESwC · 2025-10-31

**Soundness:** 4
**Presentation:** 2
**Contribution:** 3
**Rating:** 6
**Confidence:** 4

**Summary:**

The authors aim to finetune pretrained Qwen-2.5-VL models to perform code-assisted visual reasoning, executing Python code on images with operations like cropping, zooming, and general computation. They curate post-training datasets for supervised finetuning and reinforcement learning with substantial manual annotation effort. Training with SFT followed by GRPO yields a model with improved performance on a variety of benchmarks.

**Strengths:**

- Experiments are well-executed, thorough, and explore a range of design decisions.
- Dataset contribution would be substantial and fill a gap in the open-source community.
- Problem is well-motivated and timely.
- Empirical benefits are convincing on a range of benchmarks.

**Weaknesses:**

- Fully missing discussion of a substantial line of related work in code execution for visual reasoning: Visual Programming - Gupta et al and ViperGPT - Suris et al first introduced the idea of code execution at inference for visual reasoning, prior to OpenAI's 'thinking with images'; subsequent work such as Visual Sketchpad by Hu et al extended this to a larger set of image manipulations (including on some benchmarks reported here, albeit relying on proprietary models)
- If I understand correctly, GRPO-ATS = using temperature 0 for code and 1 for language, which feels like more of an implementation detail than a substantial methodological contribution. In my estimation, the paper would benefit from not putting as much focus on it, but rather presenting it as a finding with the justifications already discussed.
- The overall method essentially boils down to manually annotating data and doing post-training with SFT and GRPO, but explores the details needed to get things to work in the domain of code-assisted visual reasoning.

**Questions:**

- Perhaps I missed it, but is there a benchmark with the pretrained Qwen-2.5-VL + code execution at inference time as opposed to direct application of Qwen-2.5-VL (7B/32B) as a VLM? Is this the Qwen-2.5-VL in Table 6? The paper would benefit from clearer differentiation between inference settings.
- Will the dataset be released? Considering a substantial portion of the method relies on the manually-annotated data, this would be critical for reproducibility. The details on how the data was generated also seemed insufficient to reproduce the pipeline, like how the filtering happened in particular,  (This is my biggest concern.)

---

> ### Author Response · Authors · 2025-11-16
> **Response [1/2]**
>
> ### [Concern 1] Fully missia ng discussion of a substantial line of related work in code execution for visual reasoning
>
> Thank you for pointing out the important line of work on code execution for visual reasoning. We have conducted a comprehensive comparison with these foundational methods and many recent works. Specifically, we added comparisons with **state-of-the-art APIs** (GPT-4o, Gemini Flash 2.5, Gemini Pro 2.5, Claude 4 Sonnet, OpenAI O3) and **recent tool-use methods** (Visual Sketchpad, PyVision, ZoomEye, DyFo, Chain-of-Focus, Pixel Reasoner, DeepEyes, Mini-O3, Refocus, Visual CoT, ViperGPT, and Visual Programming).
>
> We organize these works into three categories: **(1) Training-free methods** like Visual Programming, ViperGPT, Visual Sketchpad, and PyVision that use predefined APIs and code generation without training; **(2) Code-based image manipulation methods** like Refocus that generate code for visual modifications; and **(3) Single external tool approaches** like Pixel Reasoner, DeepEyes, and Mini-O3 that treat specific operations (e.g., crop-and-zoom) as discrete tools.
>
> Our empirical evaluation across 7 benchmarks shows that Thyme achieves competitive or superior performance compared to these methods - for example, Visual Sketchpad achieves 80.3 on VStarBench using GPT-4o, while Thyme-VL-7B achieves 82.2 with a much smaller model, demonstrating the advantages of our trainable approach with a versatile toolset. **Please refer to our Response to Reviewer hmQE for the complete analysis, including detailed performance tables across all benchmarks, a thorough discussion of each category, and a comprehensive positioning of Thyme relative to these works.**
>
> Thank you for ensuring we properly acknowledge this important research lineage.
>
> ----
>
> ### [Concern 2] The paper would benefit from not putting as much focus on GRPO-ATS, but rather presenting it as a finding with the justifications already discussed.
>
> We thank the reviewer for this insightful observation. In response, we have significantly reduced the emphasis on GRPO-ATS as a “methodological novelty” in both the introduction and method sections. Instead, we now present it as an empirically motivated design choice that improves training stability and sample efficiency—consistent with the reviewer’s suggestion. The revised text clarifies that this temperature scheduling is a lightweight, task-aware practice rather than a new RL algorithm.
>
> ----
>
> ### [concern 3] The overall method essentially boils down to manually annotating data and doing post-training with SFT and GRPO
>
> The reviewer is absolutely right that the foundation of Thyme builds upon established techniques—namely supervised fine-tuning (SFT) and reinforcement learning (GRPO). However, our key contribution lies not in inventing new algorithms de novo, but in identifying, solving, and rigorously validating the specific challenges that arise when extending MLLMs to code-assisted visual reasoning—a setting where standard VLM training paradigms break down. In this sense, Thyme provides a principled recipe for training MLLMs to reliably interact with external computational environments in complex visual tasks—a capability that, as far as we know, has not been systematically achieved or benchmarked in prior work.

---

> ### Author Response · Authors · 2025-11-16
> **Response [2/2]**
>
> ### [Concern 4] Qwen-2.5-VL + code
>
> We sincerely thank the reviewer for this excellent and clarifying question.
>
> The Qwen2.5-VL-7B results reported in Table 1 (and throughout the paper) correspond to the standard inference setting—that is, the model is prompted to directly generate a textual answer without access to any code execution environment. This reflects the typical closed-loop VQA usage of off-the-shelf VLMs.
>
> To directly address the reviewer’s concern, we conducted an additional experiment in which we prompted Qwen2.5-VL-7B to generate executable code (using the same prompt template and sandbox as Thyme). However, its code execution success rate was low, primarily due to syntactic errors (e.g., invalid variable names), missing imports, and incorrect image operation logic.
>
> | Model                     | Vstar | HRBench4K | MME-Realworld | LogicVista | MathVision_MINI | MMStar | MMVet_Hard |
> |--------------------------|-------|-----------|---------------|------------|-----------------|--------|------------|
> | Qwen2.5-VL-7B (Direct)   | 76.4  | 68.8      | 58.3          | 39.8       | 27.0            | 64.5   | 52.9       |
> | Qwen2.5-VL-7B (Code)     | 62.3  | 67.75     | 57.62         | 42.95      | 26.64           | 62.6   | 53.22      |
> | Thyme-VL-7B              | 82.2  | 77.0      | 64.8          | 49.0       | 27.6            | 65.9   | 58.3       |
>
> More importantly, the impact on downstream performance differs by task type:
> 1. In reasoning-heavy tasks, the model rarely attempts to generate code (preferring direct answers), so its performance remains close to the standard baseline.
> 2. In perception-intensive tasks (e.g., high-resolution object recognition), it attempts code generation more frequently—but due to poor code quality, the sandbox often returns invalid or unhelpful results, leading to no improvement or even performance degradation compared to the direct-answer baseline.
>
> This demonstrates that merely granting a standard VLM access to a sandbox is insufficient—effective code-assisted visual reasoning requires explicit training to learn when and how to invoke tools correctly, which is precisely what our framework enables.
>
> ----
>
> ### [Concern 5] Will the dataset be released? The details on how the data was generated?
>
> We sincerely thank the reviewer for raising this important point. We fully agree that data availability is critical for reproducibility, especially given the role of our manually annotated data in the overall pipeline.
>
> We would like to reassure the reviewer that **all of our data, training code, and sandbox environment will be fully open-sourced** upon publication. Due to the large size of the SFT dataset and time constraints during the review process, we are currently releasing the **reinforcement learning (RL) dataset, sandbox environment, and the exact prompts and filtering code used in our pipeline** via an anonymous Hugging Face repository:
> 🔗 [https://huggingface.co/datasets/Openreview123/Thyme-Data](https://huggingface.co/datasets/Openreview123/Thyme-Data)
>
> In the paper, we have already described the data generation process in as much detail as possible, including releasing all functional prompts used to construct the data. Regarding the reviewer’s specific concern about filtering, we provide further clarification below:
>
> - **Non-executable code** is filtered out during generation—any code snippet that fails to run in the sandbox is discarded.
> - Responses whose **final answer does not match the execution result** of the code are removed.
> - Cases where the **model’s reasoning (thinking) contradicts its final answer** are excluded.
> - **Visual-textual inconsistencies** are also filtered: for example, if the generated code produces a blank or incorrect image (e.g., failed cropping), but the model’s reasoning claims a successful operation and provides a confident answer, such samples are discarded.
>
> We hope this additional detail addresses the reviewer’s concerns. We are committed to full transparency and will ensure all materials are publicly accessible to support reproducibility.

---

> > ### Author Response · Authors · 2025-11-26
> > **Kindly Request for Reviewer's Feedback**
> >
> > Dear Reviewer,
> >
> > Thank you so much for your time in improving our paper!
> >
> > Since the end of the rebuttal is coming soon, may we know if our response addresses your main concerns? Should you have any further advice, please let us know and we will be more than happy to engage in more discussion and improvements.

---

### Official Review · Reviewer_hmQE · 2025-11-01

**Soundness:** 2
**Presentation:** 3
**Contribution:** 3
**Rating:** 4
**Confidence:** 4

**Summary:**

The paper introduces Thyme, allowing models to autonomously generate and execute code (e.g., cropping, rotation, contrast enhancement, mathematical computations) to manipulate images and support reasoning. The authors design a two-stage training strategy: a supervised fine-tuning (SFT) on a curated ~500K-sample dataset teaching code generation and image operations, followed by a reinforcement learning (RL) phase ~ 10K images using their novel algorithm GRPO‑ATS (“Group Relative Policy Optimization with Adaptive Temperature Sampling”) to refine decision-making about code vs text generation. Thyme is evaluated across nearly 20 benchmarks spanning perception reasoning and general VQA tasks, all showing consistent improvements over baseline.

**Strengths:**

1. The training dataset size is huge (500K + 10K). Including diverse editing tools.
2. The SFT data and RL data are separate. Very clear.
3. The experiments are comprehensive. Many benchmarks are included. Signal is clear -- adding this data is helpful.

**Weaknesses:**

Main concerns are about experiment missing baselines, and missing citations.

Experiments results in Table need more baseline models. Could you also add GPT 4 and 5, GPT O3,, Gemini 2.5 Pro, Claude, [Visual Sketchpad: Sketching as a Visual Chain of Thought for Multimodal Language Models],  [Pixel Reasoner: Incentivizing Pixel-Space Reasoning with Curiosity-Driven Reinforcement Learning]...etc, as baselines?

Missing Citations on related works (because many concepts in Thyme first appeared in these related works): [Visual Sketchpad: Sketching as a Visual Chain of Thought for Multimodal Language Models], [ReFocus: Visual Editing as a Chain of Thought for Structured Image Understanding], [Visual CoT: Advancing Multi-Modal Language Models with a Comprehensive Dataset and Benchmark for Chain-of-Thought Reasoning]

**Questions:**

Please see weakness. Happy to raise score if addressed.

---

> ### Author Response · Authors · 2025-11-16
> **Response [1/2]**
>
> Thank you very much for your valuable feedback regarding the need for a more comprehensive comparison with recent baselines. We agree that this is crucial for positioning our work. We have conducted new experiments and will incorporate all of the following discussion and results into the final version of our paper.
>
> ----
>
> ### Expanded Baselines and Evaluation Setup
>
> To provide a thorough comparison, we have introduced a new set of strong baselines, including:
> *   **State-of-the-art commercial APIs:** GPT-4o, Gemini Flash 2.5, Gemini Pro 2.5, Claude 4 Sonnet, and OpenAI O3.
> *   **Recent academic works:** ZoomEye, DyFo, Chain-of-Focus, Pixel Reasoner, DeepEyes, and Mini-O3.
>
> Due to time constraints, we present results on a representative subset of benchmarks. For the API models, we evaluated their performance on **VStarBench**, **HRBench4K**, **MME-RealWorld**, **LogicVista**, **MathVision-MINI**, **MMVet-Hard**, and **MMStar**. This selection covers three perception, two reasoning, and two general-purpose benchmarks. We compare Thyme with these methods both conceptually and in terms of performance. For recent methods like DeepEyes and Visual Sketchpad, we identify common benchmarks for a direct comparison and provide further conceptual analysis.
>
> ----
>
> ### Conceptual Comparison with Prior Work
>
> We categorize previous tool-use methods into three groups to clarify their distinct approaches and highlight the novelty of Thyme.
>
> **1. Training-free methods:** These methods use LLMs to generate code or plans to call predefined visual APIs.
> *   **ViperGPT [8]** and **VisProg [9]** predefine a set of visual APIs (e.g., detection, segmentation) and prompt a model to generate code to orchestrate these tools for complex tasks. These works function more like agent-based systems that compose multiple expert models.
> *   **Visual Sketchpad [1]** enables an off-the-shelf MLLM (like GPT-4o) to actively draw on a visual "sketchpad" (e.g., adding boxes, lines, masks) and use these visual artifacts for subsequent reasoning.
> *   **PyVision [7]** is a more recent work that prompts a model to generate PIL (Python Imaging Library) code for image manipulations.
> The performance of these methods is highly dependent on the quality of the user's query and the predefined rules, as they do not undergo task-specific training.
>
> **2. Code-based image manipulation methods:** These methods generate code to perform visual modifications.
> *   **Refocus [5]** prompts a model to generate code to draw boxes, masks, or highlights, focusing perception on critical regions. However, the potential of code generation is not fully realized, and drawing boxes is insufficient for resolving high-resolution perception challenges, such as magnifying small objects.
>
> **3. Single external tool approaches:** These methods integrate a specific operation as a single, external tool.
> *   **Pixel Reasoner [2]**, **DeepEyes [3]**, **Mini-O3 [4]**, and **Visual CoT [6]** all treat crop-and-zoom as a discrete external tool to enhance the model's perceptual capabilities. Compared to Thyme, their toolsets are limited, which restricts their flexibility.
>
> ----
>
> ### Performance Comparison and Analysis
>
>
> **Analysis:**
>
> First, we observe that state-of-the-art open-source models like DeepEyes and Thyme achieve performance on perception tasks that is largely comparable to Gemini Flash 2.5. (We note that the VStarBench benchmark includes a training set, and the suboptimal performance of most closed-source models may be because they have not been exposed to similar data). However, a performance gap remains when compared to top-tier models like Gemini Pro 2.5 and O3.
>
> Compared to training-free methods such as Visual Sketchpad and PyVision-GPT-4.1, Thyme demonstrates a clear performance advantage. When compared to methods that focus solely on crop-and-zoom operations like DeepEyes and Mini-O3, Thyme's perception performance is competitive, and it demonstrates significant advantages on reasoning tasks, showcasing its greater flexibility. These results underscore the benefits of Thyme's versatile toolset. We will integrate this detailed comparison and analysis into our final manuscript (Section F in the Appendix).

---

> ### Author Response · Authors · 2025-11-16
> **Response [2/2]**
>
> The evaluation results of these methods and the commercial APIs are summarized in the table below.
>
> | Method                 | VStarBench | HRBench4K | MME-RealWorld | LogicVista | MathVision-MINI | MMStar  | MMVet-Hard |
> | ---------------------- | ---------- | --------- | ------------- | ---------- | --------------- | ------- | ---------- |
> | *Tool-Use Methods*     |            |           |               |            |                 |         |            |
> | Visual Sketchpad       | 80.3       | -         | -             | -          | -               | -       | -          |
> | PyVision-GPT-4.1       | 75.9       | -         | -             | -          | -               | -       | -          |
> | ZoomEye                | 90.6       | 69.6      | -             | -          | -               | -       | -          |
> | DyFo                   | 81.2       | -         | -             | -          | -               | -       | -          |
> | Chain-of-Focus         | 88.0       | -         | -             | -          | -               | -       | -          |
> | Pixel Reasoner         | 86.3       | 74.0      | 64.4          | -          | -               | -       | -          |
> | DeepEyes               | 83.3       | 73.2      | 64.0          | 47.7       | 26.6            | 63.87   | 53.23      |
> | Mini-O3                | 88.2       | 77.5      | 65.5          | -          | -               | -       | -          |
> | *Our Methods*          |            |           |               |            |                 |         |            |
> | Qwen2.5-VL-7B (Direct) | 76.4       | 68.8      | 58.3          | 39.8       | 27.0            | 64.5    | 52.9       |
> | **Thyme-VL-7B**        | **82.2**   | **77.0**  | **64.8**      | **49.0**   | **27.6**        | **65.9**| **58.3**   |
> | *Commercial APIs*      |            |           |               |            |                 |         |            |
> | GPT-4o                 | 67.5       | 65.0      | 62.8          | 53.2       | 36.5            | 65.7    | 58.3       |
> | Gemini Flash 2.5       | 73.82      | 79.25     | 60.9          | 47.65      | 35.19           | 70.0    | 73.19      |
> | Gemini Pro 2.5         | 79.05      | 83.87     | 71.27         | 46.3       | 33.55           | 72.0    | 76.97      |
> | Claude 4 Sonnet        | 56.4       | 49.37     | 59.87         | 63.53      | 48.0            | 54.8    | 55.45      |
> | O3                     | 95.7       | 84.82     | 72.48         | 67.33      | 63.81           | 75.6    | 67.24      |
>
>
> ### References
>
> [1] Hu, Yushi, et al. "Visual sketchpad: Sketching as a visual chain of thought for multimodal language models." Advances in Neural Information Processing Systems 37 (2024): 139348-139379.
>
> [2] Su, Alex, et al. "Pixel reasoner: Incentivizing pixel-space reasoning with curiosity-driven reinforcement learning." arXiv preprint arXiv:2505.15966 (2025).
>
> [3] Zheng, Ziwei, et al. "DeepEyes: Incentivizing" Thinking with Images" via Reinforcement Learning." arXiv preprint arXiv:2505.14362 (2025).
>
> [4] Lai, Xin, et al. "Mini-o3: Scaling up reasoning patterns and interaction turns for visual search." arXiv preprint arXiv:2509.07969 (2025).
>
> [5] Fu, Xingyu, et al. "Refocus: Visual editing as a chain of thought for structured image understanding." arXiv preprint arXiv:2501.05452 (2025).
>
> [6] Shao, Hao, et al. "Visual cot: Advancing multi-modal language models with a comprehensive dataset and benchmark for chain-of-thought reasoning." Advances in Neural Information Processing Systems 37 (2024): 8612-8642.
>
> [7] Zhao, Shitian, et al. "Pyvision: Agentic vision with dynamic tooling." arXiv preprint arXiv:2507.07998 (2025).
>
> [8] Surís, Dídac, Sachit Menon, and Carl Vondrick. "Vipergpt: Visual inference via python execution for reasoning." Proceedings of the IEEE/CVF international conference on computer vision. 2023.
>
> [9] Gupta, Tanmay, and Aniruddha Kembhavi. "Visual programming: Compositional visual reasoning without training." Proceedings of the IEEE/CVF conference on computer vision and pattern recognition. 2023.
>
> ***
>
> Once again, we thank you for your constructive comments, which have helped us significantly improve the paper.

---

> > ### Author Response · Authors · 2025-11-26
> > **Kindly Request for Reviewer's Feedback**
> >
> > Dear Reviewer,
> >
> > Thank you so much for your time in improving our paper!
> >
> > Since the end of the rebuttal is coming soon, may we know if our response addresses your main concerns? Should you have any further advice, please let us know and we will be more than happy to engage in more discussion and improvements.

---

### Meta-Review · Area_Chair_NNnA · 2026-01-11

**Summary:**

The main concerns from the reviewers are following:

- Reviewer **hmQE**:
  - **W1**: Main concerns are about experiment missing baselines, and missing citations.

- Reviewer **ESwC**:
  - **W1**: Missing discussion of a substantial line of related work in code execution for visual reasoning.
  - **W2**: Reducing the discussions on GRPO-ATS in the main paper.
  - **W3**: The proposed approach is based on established post-training pipeline, e.g. SFT and GRPO.
  - **W4**: Providing results of an additional baseline.
  - **W5**: Ensuring open-source the related resources.

- Reviewer **pa3i**:
  - **W1**: Lacking ablations on GRPO-ATS.
  - **W2**: Ensuring open-source the related resources.

**Reviewer Concerns:**

- **Concerns addressed in the rebuttal:**
  - The authors have provided additional discussions, experimental results, and clarifications to address the issues of missing citations, lacking experimental results, ensuring open-source.

- **Concerns remained outstanding:**
  - I agree with **W3** of Reviewer **ESwC** that the contribution of the paper is somehow engineering flavored.

**Reviewer Scores:**

As discussed in the Reviewer Concerns, most points are indeed addressed by the author responses. I think some reviewers will indeed raise their scores (e.g. Reviewer **pa3i**). Considering the remaining issue that the paper lacks strong technical contribution, I think the paper has done a good in job proposing a well-engineered system for training the think-with-image ability of open-source MLLMs, making it a good-value resource in the area.

---

### Decision · Program_Chairs · 2026-01-26

Accept (Poster)